# Caveolin-1 mediates cellular distribution of HER2 and affects trastuzumab binding and therapeutic efficacy

Patrícia M.R. Pereira[1], Sai Kiran Sharma[1], Lukas M. Carter[1], Kimberly J. Edwards[1], Jacob Pourat[1], Ashwin Ragupathi[1], Yelena Y. Janjigian[2], Jeremy C. Durack[1,3] & Jason S. Lewis [1,3,4,5,6]

Human epidermal growth factor receptor 2 (HER2) gene amplification and/or protein overexpression in tumors is a prerequisite for initiation of trastuzumab therapy. Although HER2 is a cell membrane receptor, differential rates of endocytosis and recycling engender a dynamic surface pool of HER2. Since trastuzumab must bind to the extracellular domain of HER2, a depressed HER2 surface pool hinders binding. Using in vivo biological models and cultures of fresh human tumors, we find that the caveolin-1 (CAV1) protein is involved in HER2 cell membrane dynamics within the context of receptor endocytosis. The translational significance of this finding is highlighted by our observation that temporal CAV1 depletion with lovastatin increases HER2 half-life and availability at the cell membrane resulting in improved trastuzumab binding and therapy against HER2-positive tumors. These data show the important role that CAV1 plays in the effectiveness of trastuzumab to target HER2-positive tumors.

[1] Department of Radiology, Memorial Sloan Kettering Cancer Center, New York, NY 10065, USA. [2] Department of Medicine, Memorial Sloan Kettering Cancer Center and Weill Cornell Medical College, New York, NY 10065, USA. [3] Department of Radiology, Weill Cornell Medical College, New York, NY 10065, USA. [4] Molecular Pharmacology Program, Memorial Sloan Kettering Cancer Center, New York, NY 10065, USA. [5] Department of Pharmacology, Weill Cornell Medical College, New York, NY 10065, USA. [6] Radiochemistry and Molecular Imaging Probes Core, Memorial Sloan Kettering Cancer Center, New York, NY 10065, USA. Correspondence and requests for materials should be addressed to J.S.L. (email: lewisj2@mskcc.org)

Unrestrained activation of human epidermal growth factor receptor 2 (HER2) contributes to aberrant tumor growth; and HER2 gene amplification, messenger RNA or protein overexpression, has been observed in patients with breast or ovarian cancer[1]. HER2 overexpression has also been reported in patients with gastric cancer, bladder carcinomas, gallbladder, and extrahepatic cholangiocarcinomas[2]. HER2 has no known ligand, but remains the most preferred dimerization partner to potentiate downstream oncogenic signaling by members of the HER family. Prior to the development of targeted anti-HER2 therapy, patients with HER2-positive tumors demonstrated reduced disease-free survival compared to patients whose tumors expressed low levels of HER2[3]. These findings established HER2 as a therapeutic target and a tumor biomarker. Over the past two decades, clinical evidence has unequivocally demonstrated that the inhibition of this oncogene improves treatment outcomes, and has led to the emergence of several effective anti-HER2 therapies[4].

Among these agents, anti-HER2 therapeutic antibodies (e.g., trastuzumab and pertuzumab), antibody-drug conjugates (ADCs, e.g., trastuzumab emtansine; TDM1), and trastuzumab imaging agents (when radio- or fluorescently-labeled[5–8]) have changed the prognosis of both breast and gastric cancer patients. However, heterogeneity in HER2 expression or equivocal HER2 status warrants attention in trastuzumab-based imaging and therapeutic strategies[9–13]. A lack of correlation between histologic HER2-positivity and tumor uptake of, e.g., zirconium-89 ($^{89}$Zr)-labeled trastuzumab has been observed in patients with breast cancer[7,14]. These results suggest that determination of overall amplification and/or overexpression of HER2 alone are insufficient to predict response to treatment with trastuzumab.

Clinically, the anti-tumor activity of trastuzumab is attributed to more than a single mechanism of action. Direct action of the antibody is premised on receptor downregulation and subsequent alterations to intracellular signaling including attenuation of downstream pro-tumorigenic cell signaling, inhibition of HER2 shedding, and inhibition of tumor angiogenesis. On the other hand, indirect action due to activation of an immune response via antibody dependent cell-mediated cytotoxicity (ADCC) has also been proposed as a mechanism of action for this drug[15–17]. Trastuzumab binding to cancer cells is highly dependent on the availability of HER2 at the cell membrane. The current status of patient selection for trastuzumab therapy is based on HER2-positivity using DNA- and protein-based assays[18]. However, these assays could overestimate HER2-positivity, as some of the stained antigen may be intracellular and, therefore, unavailable to engage trastuzumab at the tumor cell surface. This would translate as minimal benefit to such patients from trastuzumab-based therapy since the antibody can only target HER2 available at the cell membrane.

Notably, cell-surface receptors involved in tumor development are characterized by abnormal trafficking from the cell membrane to intracellular compartments[19,20]. Distinct from HER2, endocytosis of the other members of the HER family occurs after ligand binding[20]. Although HER2 has no known ligand, the open conformation of the extracellular domain contributes to the dynamics of the HER2 surface pool[21,22]. The localization of HER2 at the membrane is a heterogeneous and dynamic process[19,23,24] governed by differential rates of endocytosis and recycling[20,24,25]. In addition to cell membrane expression, HER2 localizes in the cytoplasm[26] and nucleus[27].

Several studies have demonstrated that at the cell membrane, HER2 is localized in caveolae domains[28]. Caveolae are caveolin-1 (CAV1) enriched subdomains of the plasma membrane, which are deregulated in cancer cells and contain a high content of cholesterol and sphingolipids[29,30]. Others have demonstrated that HER2 co-localizes with the lipid-raft marker GM1[31] and interacts with the lipid-raft associated oncoprotein mucin-1-c[32]. After binding to the extracellular domain of HER2, the tumor suppressor opioid binding protein/cell adhesion molecule like (OPCML) induces HER2 degradation through the caveolae-associated endosomal pathway[32]. Importantly, the depletion of membrane cholesterol results in HER2 confinement at the cell membrane[33].

Intrinsic defects in the endocytic machinery responsible for HER2 downregulation correlate with unresponsiveness to trastuzumab therapy[34]. It is suggested that caveolae-mediated endocytosis is involved in resistance to treatment with TDM1 and trastuzumab[35,36]. We hypothesize that modulation of the caveolae membrane domains could promote stability of HER2 at the cell membrane and improve binding of trastuzumab to target cells. In this study, we demonstrate how CAV1 can influence HER2 expression and stability at the cell membrane.

## Results

**HER2 interacts with CAV1.** To address cell-surface HER2 stability in CAV1-positive vs. CAV1 negative cells, we first explored protein–protein associations between HER2 and CAV1 in a panel of cancer cell lines (Fig. 1a and Supplementary Fig. 1a, b). Mining the Cancer Cell Line Encyclopedia database[37] for breast, gastric and bladder cancer lines revealed that protein expression of HER2 is inversely correlated with that of CAV1, bearing a Spearman's rank correlation coefficient ($r$) of $-0.54$ ($P < 0.0001$, Spearman's correlation). Western blot analysis of a panel of 14 cell lines supported the inverse correlation between HER2 and CAV1 protein expression ($r = -0.58$, $P = 0.03$, Spearman's correlation). UMUC14 bladder, NCIN87 gastric, BT474 and SKBR3 breast cancer cell lines exhibited the highest ratio of HER2-to-CAV1 protein levels (Fig. 1a) and were chosen for use in subsequent in vitro and in vivo experiments. HER2–CAV1 protein associations were also correlated using the STRING database, which allows for the determination of protein–protein interactions based on experimental and computationally predicted associations. This analysis yielded a high-confidence association value (score ≥ 0.700, Supplementary Fig. 1c). The network generated for HER2 and CAV1 proteins using the STRING database revealed that CAV1 has a negative molecular action on HER2. Altogether, these findings support a seesaw effect between CAV1 and HER2 proteins.

**CAV1 contributes to cell membrane HER2 density.** Given that CAV1 associates with HER2 and that CAV1 expression in malignancies may be clinically significant for cancer diagnosis[29,30], we sought to determine if CAV1 expression is correlated with HER2 density at the cell membrane. We observed that HER2 does not exhibit predominant membrane localization in all cell clusters of NCIN87 (HER2-positive gastric cancer cell line[8]) tumor xenografts (Fig. 1b). These observations are consistent with recent reports of HER2-positive gastric tumors having different HER2 densities at the cell membrane[38]. In cancer cells where CAV1 is absent, HER2 is exclusively present at the cell membrane (Fig. 1c highlighted by the dashed white circles 1 and 3). On the other hand, cancer cells expressing CAV1 exhibit reduced HER2 staining at the cell membrane (Fig. 1c highlighted by the dashed white circles 2 and 4). Consistent with these findings, immunofluorescence staining of HER2-positive tumor samples obtained from gastric cancer patients revealed that cells with high expression of CAV1 have low membrane staining of HER2 (Fig. 1d). In HER2-positive tumor samples containing low expression levels of CAV1, HER2 exhibits predominant membrane staining. Altogether, the immunofluorescence analyses of

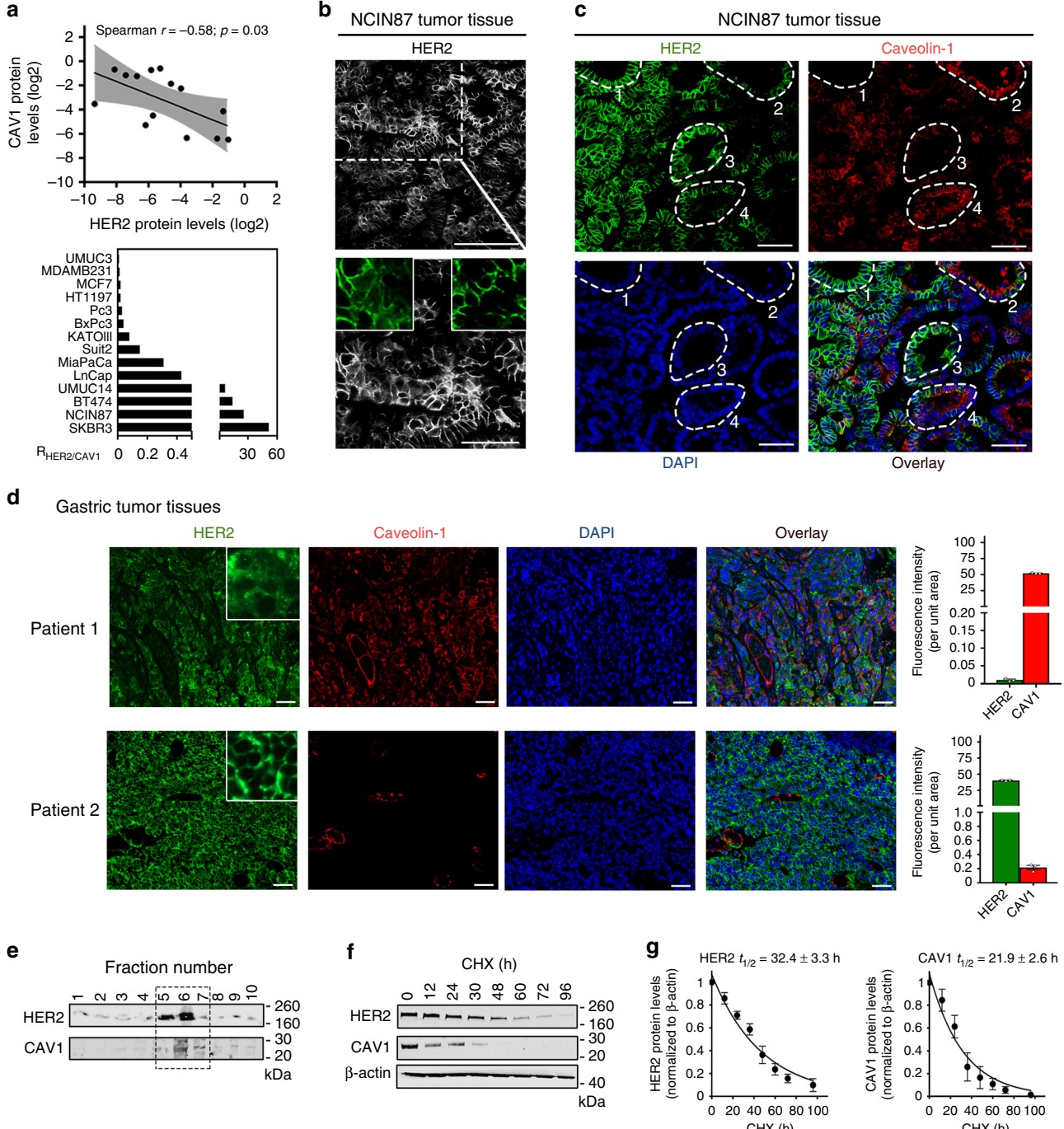

**Fig. 1** Seesaw effect between CAV1 and HER2 proteins. **a** Association between CAV1 and HER2 in gastric (NCIN87, KATOIII), bladder (HT1197, UMUC14, UMUC3), breast (MDAMB231, MCF-7, BT474, SKBR3), pancreatic (BxPc3, MiaPaCa, Suit2) and prostate (LnCap, Pc3) cancer cells. The level of CAV1 ($y$-axis) and HER2 ($x$-axis) are expressed as the ratio of protein/β-actin control, as determined by western blot analysis (upper panel). The non-parametric one-tailed Spearman test was used to determine the correlation coefficients. $R_{HER2/CAV1}$, ratio of HER2-to-CAV1 protein levels as determined by western blot analysis from total extracts (lower panel). **b** Confocal images of immunofluorescence staining of HER2 in s.c. NCIN87 gastric tumors from athymic nude mice, showing that HER2 does not exhibit a predominant membrane staining. Scale bars: 100 μm and 50 μm (inset). **c** Confocal images of immunofluorescence staining of HER2 and CAV1 in s.c. NCIN87 gastric tumors from athymic nude mice. HER2 presence at the cell membrane in cells where CAV1 is absent (cell clusters 1 and 3) and HER2 almost absent at the cell membrane in cells where CAV1 is present (cell clusters 2 and 4). Scale bars: 50 μm. **d** Confocal images and quantification (mean ± S.E.M, $n = 3$) of immunofluorescence staining of HER2 and CAV1 in human HER2-positive gastric tumors containing high (Patient 1) and low (Patient 2) expression of CAV1. HER2 predominant presence at the cell membrane in CAV1-negative tumors (Patient 2). Scale bars: 50 μm. **e** Western blot analysis of the distribution of HER2 and CAV1 in caveolae fractions isolated from NCIN87 cells. **f** Western blot of HER2 and CAV1 in the total lysates of NCIN87 cells after blocking protein synthesis with 80 μg mL$^{-1}$ CHX for 0, 12, 24, 30, 48, 60, 72, and 96 h. CHX cyclohexamide. **g** HER2 and CAV1 have similar half-lives. Half-lives calculated after western blot analysis of Fig. 1f. Density of western blot bands was quantified by scanning densitometry with ImageJ software. Half-lives were calculated as the time required for protein decrease to 50% of its initial level (mean ± S.E.M, $n = 4$)

preclinical and clinical samples indicate that CAV1 expression plays a role in HER2 antigen density at the cell membrane.

Additional subcellular fractionation experiments confirmed that HER2 is present, with CAV1, in the same lipid-rich fractions (Fig. 1e). HER2 and CAV1 half-lives were estimated in cells treated with an inhibitor of protein biosynthesis, cycloheximide, CHX (Fig. 1f, g and Supplementary Fig. 2). The actin-normalized HER2 or CAV1 protein expression levels (relative to t = 0) were plotted over time, and protein half-life was estimated from a monophasic exponential fit using GraphPad Prism (Fig. 1g). The estimated half-life of HER2 was similar to CAV1 in NCIN87 gastric, BT474 and SKBR3 breast cancer cell lines. Collectively, these results support the association between CAV1 and HER2.

**Depletion of CAV1 improves trastuzumab binding.** The inverse correlation between CAV1 and HER2 expression led us to

investigate if depletion of CAV1 would result in improved HER2 stability at the cell membrane. To this end, we first used a pool of three target-specific 20–25 nucleotides (nt) small-interfering RNA (siRNA) to knockdown CAV1 protein (Supplementary Fig. 3a). Cell-surface biotinylation experiments were performed to analyze HER2 expression and localization in the plasma membrane. siRNA-mediated knockdown of CAV1 slightly increased (1.4 ± 0.2, mean ± S.E.M, n = 3) HER2 at the cell membrane for NCIN87 cells (Fig. 2a) and UMUC14 cells (Supplementary Fig. 3b), without alteration of total HER2 protein levels. In contrast, when we used clustered regularly interspaced short palindromic repeats (CRISPR) activation plasmid to increase CAV1, we found that forced CAV1 overexpression promotes loss of HER2 at the cell membrane (Fig. 2b and Supplementary Fig. 3c). While CAV1 depletion resulted in a slight increase in cell-surface HER2, a more pronounced effect of knocking down CAV1 was demonstrated by the extended half-life

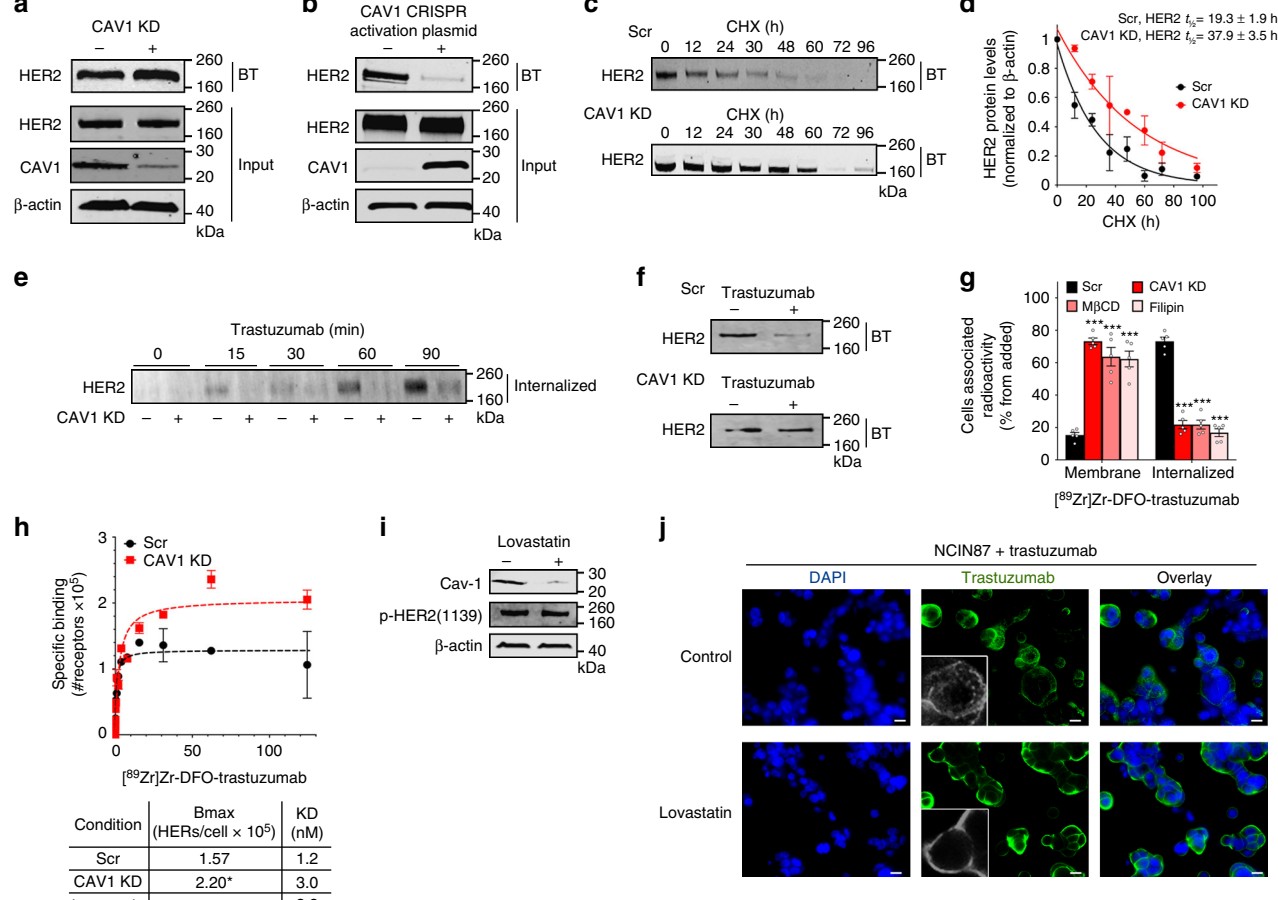

**Fig. 2** In vitro CAV1 depletion increases HER2 stability at the cell membrane. **a** Western blot of biotinylated cell surface-associated HER2 along with HER2 and CAV1 in total lysates of NCIN87 cells transfected with CAV1 siRNA or scrambled control siRNA constructs or **b** transfected with CAV1 CRISPR activation plasmid or control CRISPR activation plasmid. HER2 protein levels increase at the cell surface after CAV1 depletion (1.4 ± 0.2, mean ± S.E.M, n = 3). BT biotinylation. **c** Western blot of biotinylated cell surface-associated HER2 in NCIN87 cells transfected with CAV1 siRNA or scrambled control siRNA constructs and blocking protein synthesis with 80 μg mL$^{-1}$ CHX. CHX cyclohexamide. **d** Half-lives of cell surface-associated HER2 (mean ± S.E.M, *P < 0.05 based on a Student's t-test, n = 5). **e, f** Western blot analysis of internalized HER2 (**e**) or cell surface–associated HER2 (**f**) on NCIN87 cells transfected with either control or CAV1 siRNA and treated with 1 μM trastuzumab. **g** Membrane-bound and internalized [$^{89}$Zr]Zr-DFO-trastuzumab before and after CAV1 knockdown or CAV1 depletion with methyl-β-cyclodextrin (MβCD) or filipin in NCIN87 cancer cells (bars, n = 5, mean ± S.E.M, ***P < 0.001 based on a Student's t-test). **h** NCIN87 cells transfected with CAV1 siRNA or scrambled control siRNA constructs were incubated with [$^{89}$Zr]Zr-DFO-trastuzumab (0 to 125 μM) for 3 h at 4 °C (upper panel). Specific binding of [$^{89}$Zr]Zr-DFO-trastuzumab (black circles, red squares) and non-linear regression curve fit (dotted lines). Data are presented as mean ± S.E.M, n = 3. Binding parameters (lower panel) of [$^{89}$Zr]Zr-DFO-trastuzumab to NCIN87 cells scrambled control, treated with CAV1 siRNA or with lovastatin (*P < 0.05 based on a Student's t-test, n = 5). **i** Western blot of CAV1 and phosphorylated HER2 proteins from total extracts of NCIN87 cells incubated with 25 μM of lovastatin for 4 h. β-actin was used as a loading control. **j** Confocal images of immunofluorescence of HER2 in NCIN87 cells incubated with 1 μM trastuzumab for 90 min in the presence and absence of lovastatin. Scale bars: 20 μm

($P < 0.05$, Student's $t$-test) of HER2 at the cell membrane (Fig. 2c, d and Supplementary Fig. 3d, e) in NCIN87, UMUC14, and BT474 cancer cells. Knockdown of CAV1 in HER2-positive SKBR3 breast cancer cell line, did not impact the HER2 cell-surface half-life (Supplementary Fig. 3f) plausibly due to the low levels of caveolae in this cell line[31] (Supplementary Fig. 1a). In direct comparison, the membrane-bound HER2 half-life was 2.2-, 1.5-, and 2.0-fold higher in CAV1-depleted NCIN87, UMUC14 and BT474 cells, respectively.

Following trastuzumab binding, HER2 is internalized from the plasma membrane and accumulates in sorting endosomes. Thereafter, it can either recycle back to the cell surface or be degraded in the lysosomes[39]. We sought to investigate if decreased trastuzumab-mediated HER2 internalization could be achieved by a temporal and acute depletion of CAV1. To this end, trastuzumab was allowed to bind HER2 on NCIN87 cells treated with a scrambled (Scr) siRNA (control) vs. cells treated with a CAV1-depleting siRNA. The cells were allowed to undergo trastuzumab-mediated endocytosis for at least 90 min after surface biotinylation, and the internalized fraction of HER2 was evaluated via immunoblotting (Fig. 2e and Supplementary Fig. 4a). Incubation with trastuzumab depresses the surface pool of HER2 (Fig. 2f) as the receptor is internalized in a time-dependent manner. Notably, compared to scr cells, a marked decrease in HER2 endocytosis was observed in CAV1-depleted NCIN87 cells. Cellular fractionation of CAV1-depleted cancer cells treated with [89]Zr-labeled trastuzumab revealed a significant increase ($P < 0.001$, Student's $t$-test) in membrane-associated radioactivity (Fig. 2g and Supplementary Fig. 4b). Increased membrane-associated [89]Zr-labeled trastuzumab was also observed upon depletion of cholesterol using methyl-β-cyclodextrin (MβCD) and filipin (Fig. 2g and Supplementary Fig. 4b). Next, the binding profile of [89]Zr-labeled trastuzumab in competitive radioligand saturation-binding assays confirmed that CAV1 knockdown increased HER2 density on the cell membrane ($B_{max}$) (Fig. 2h and Supplementary Fig. 4c). Altogether, these data indicate that CAV1 depletion stabilizes HER2 at the cell membrane and improves the tumor avidity for trastuzumab on HER2-positive cancer cells.

**CAV1 can be downregulated in vivo by lovastatin.** Premised on our in vitro findings, we expected that HER2-positive tumors containing high levels of CAV1 would have a low surface pool of HER2. This in turn would impact the ability of trastuzumab to target and treat HER2-positive cancer cells. Therefore, we sought to investigate whether CAV1 depletion might result in an increased uptake of trastuzumab in tumor xenografts and human tumor samples. However, to date, there are no commercially available CAV1-specific modulators. Nevertheless, protein levels of CAV1 can be reduced by disruption of cholesterol-rich membrane domains using cholesterol-lowering drugs[30,39].

Lovastatin is one such commercially available drug that is prescribed worldwide to normalize lipid content in individuals having high blood cholesterol levels. In our study, lovastatin was used as a pharmacological modulator of CAV1. Previous studies have reported that 25 μM of lovastatin reduces the protein expression of CAV1 in vitro[40]. In our studies, the active form of lovastatin effectively depleted CAV1 expression in NCIN87 cells without altering phosphorylated HER2 levels (Fig. 2i). Thus, we further explored the potential role of lovastatin as a CAV1 modulator in the context of trastuzumab binding to NCIN87 cells. A potent role for the active form of lovastatin in increasing trastuzumab binding to cancer cells was also demonstrated via competitive radioligand saturation-binding assays (Fig. 2h and Supplementary Fig. 4c). These results seem to indicate that the

increased binding of trastuzumab to HER2-positive cancer cells after treatment with lovastatin is a result of an overall increase in the binding between trastuzumab and HER2. Using fluorescence microscopy, trastuzumab was found not only on the surface of cancer cells, but also in intracellular vesicles (Fig. 2j and Supplementary Movie 1). However, in the presence of lovastatin, trastuzumab exhibited predominant membrane staining (Fig. 2j and Supplementary Movie 2). Lovastatin's inhibition of the enzyme 3-hydroxyl-3-methylgutarylcoenzyme A (HMG-CoA) reductase (HMGCR) is known to result in the decrease of mevalonate and its downstream products (e.g., cholesterol). In the presence of lovastatin, cells exhibited an increase in membrane-associated [89]Zr-labeled trastuzumab, an effect that is rescued by the addition of mevalonate to the cell culture (Fig. 3a and Supplementary Fig. 4d). Additional studies demonstrated that an increment in membrane cholesterol content has the opposite effect (Fig. 3a and Supplementary Fig. 4d). In sum, our findings suggest that lovastatin may be able to pharmacologically modulate HER2 endocytosis through a CAV1-mediated mechanism.

Further to these findings, we explored the efficiency of lovastatin to deplete CAV1 in NCIN87 xenografts. Two doses of lovastatin (8.3 mg kg$^{-1}$ of mice), administered within 12 h of each other, was established as the most efficient treatment regimen that yielded a significant reduction in the total levels of CAV1 protein (Fig. 3b). Since the extracellular domain of HER2 is known to shed in vivo, we evaluated the impact of lovastatin treatment on this phenomenon. Specifically, treatment with lovastatin reduced HER2 shedding by approximately 50% in vitro using NCIN87 cells vs. 70% in NCIN87 xenografts (Fig. 3c). HER2 localization at the plasma membrane was increased in NCIN87 xenografts treated with lovastatin vs. saline-treated control mice (Fig. 3d–f and Supplementary Fig. 5). No further reduction of CAV1 concomitant with an increase in cell surface HER2 in tumors could be observed when mice were treated with lovastatin for longer than 60 h. Collectively, these results demonstrate that CAV1 can be temporally modulated in vitro and in vivo via pharmacological treatment with a statin.

**CAV1 depletion increases tumor avidity for trastuzumab.** To determine the impact of lovastatin-mediated CAV1 depletion on trastuzumab's binding to HER2-positive tumors, a pilot study was performed using separate bilateral xenografts of UMUC14 bladder cancer and NCIN87 gastric cancer (Supplementary Fig. 6). The active form of lovastatin (0.44 mg kg$^{-1}$) was injected into the tumor on the left flank while the right-flank tumor was used as a control (CT). To noninvasively monitor the effect of this treatment, mice were injected intravenously (i.v.) with [89]Zr-labeled trastuzumab at 4 h post-administration of lovastatin. An additional intratumoral injection of lovastatin was performed simultaneously with the injection of [89]Zr-labeled trastuzumab. Notably, there was a significant difference in the uptake of [89]Zr-labeled trastuzumab between the lovastatin vs. CT tumors (Supplementary Fig. 6a, b). At 48 h post-injection of [89]Zr-labeled trastuzumab, NCIN87 CT tumors had an uptake of 19.7 ± 11.8% ID g$^{-1}$, while tumors treated with lovastatin yielded an uptake of 41.6 ± 10.7 % ID g$^{-1}$. Similarly, the uptake of the radiotracer in UMUC14 xenografts was 13.6 ± 3.6% ID g$^{-1}$ in CT tumors vs. 19.9 ± 3.6% ID g$^{-1}$ in lovastatin-treated tumors (Supplementary Fig. 6c, d). The higher accumulation of trastuzumab in NCIN87 tumors vs. UMUC14 is a result of higher HER2 expression in NCIN87 cells (Supplementary Fig. 1a).

Encouraged by the results of this pilot study, we performed in vivo studies in mice bearing unilateral NCIN87 tumors (Fig. 4a and Supplementary Fig. 7). Since lovastatin is clinically prescribed

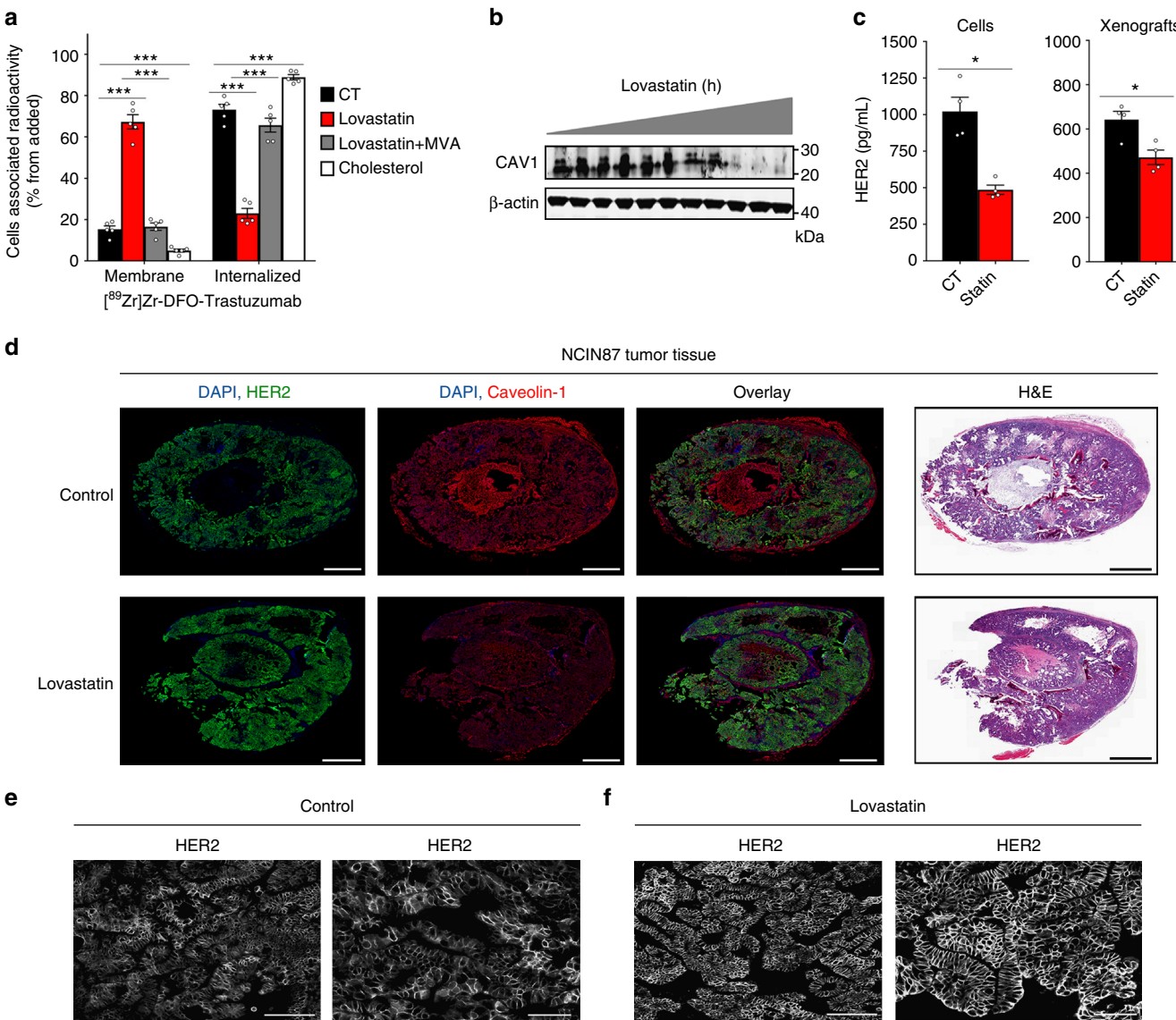

**Fig. 3** In vivo CAV1 modulation with statin improves HER2 stability at the cell membrane. **a** Membrane-bound and internalized [$^{89}$Zr]Zr-DFO-trastuzumab before and after CAV1 depletion with 25 µM of lovastatin for 4 h in NCIN87 cancer cells. Increase in membrane-bound trastuzumab is rescued when cells are treated with lovastatin in the presence of 200 µM mevalonic acid (MVA). Treatment of NCIN87 cancer cells with a MβCD solution saturated with cholesterol increases [$^{89}$Zr]Zr-DFO-trastuzumab internalization (bars, $n = 5$, mean ± S.E.M, **$P < 0.01$, ***$P < 0.001$ based on a Student's $t$-test, $n = 4$). **b** Western blot of CAV1 in the total lysates of NCIN87 s.c. tumors from athymic nude mice. Lovastatin (8.3 mg kg$^{-1}$ of mice) was orally administrated twice with an interval of 12 h between each administration. Tumor lysates were prepared between 0 and 48 h after the second dose of lovastatin and analyzed by western blot. **c** Treatment with lovastatin decreases HER2 shedding as determined by ELISA experiments (bars, $n = 4$, mean ± S.E.M, *$P < 0.05$ based on a Student's $t$-test). **d** Fluorescence staining (HER2 and CAV1) and H&E stain of tissue sections of s.c. NCIN87 gastric tumors from athymic nude mice. Lovastatin (8.3 mg kg$^{-1}$ of mice) was orally administrated twice with an interval of 12 h between each administration. Tumor sections were prepared at 48 h after the first dose of lovastatin. Scale bars: 100 µm. **e**, **f** Lovastatin increases cell membrane HER2 in NCIN87 tumors. Confocal images of immunofluorescence staining of HER2 in s.c. NCIN87 gastric tumors from athymic nude mice, non-treated (**e**) or treated with lovastatin (**f**). Images are two representative sections of two different xenografts. Lovastatin (8.3 mg kg$^{-1}$ of mice) was orally administrated twice with an interval of 12 h between each administration. Tumor sections were prepared at 48 h after the first dose of lovastatin. Scale bars: 100 µm (left) and 50 µm (right)

as an orally administered formulation, further studies were performed using oral administration of lovastatin 12 h prior and at the same time of trastuzumab injection. The 12 h window was chosen since immunoblotting and immunofluorescence data demonstrated that CAV1 was significantly reduced in the tumor with a concomitant increase in HER2 at the cell membrane at this time point (Fig. 3). Control mice were orally administered saline instead of lovastatin. Longitudinal PET imaging of these mice revealed tumors that were clearly delineated as early as 4 h post-injection of trastuzumab and yielded high contrast images at 48 h

post-injection of the radiotracer (Fig. 4a). Ex vivo biodistribution studies further validated our findings from PET imaging (Fig. 4b). The saline-treated cohort yielded a radiopharmacologic profile typical of $^{89}$Zr-labeled trastuzumab, which shows a gradual accretion into the HER2-positive tumors (2.6 ± 1.3% ID g$^{-1}$ at 4 h to 17.5 ± 6.9% ID g$^{-1}$ at 48 h). On the other hand, xenograft mice treated with lovastatin yielded a remarkably high tumor uptake of 9.1 ± 4.6% ID g$^{-1}$ as early as 4 h post-injection of $^{89}$Zr-labeled trastuzumab, which rose to 27.1 ± 7.6 at 48 h. High tumor-to-background ratios were achieved in mice pretreated

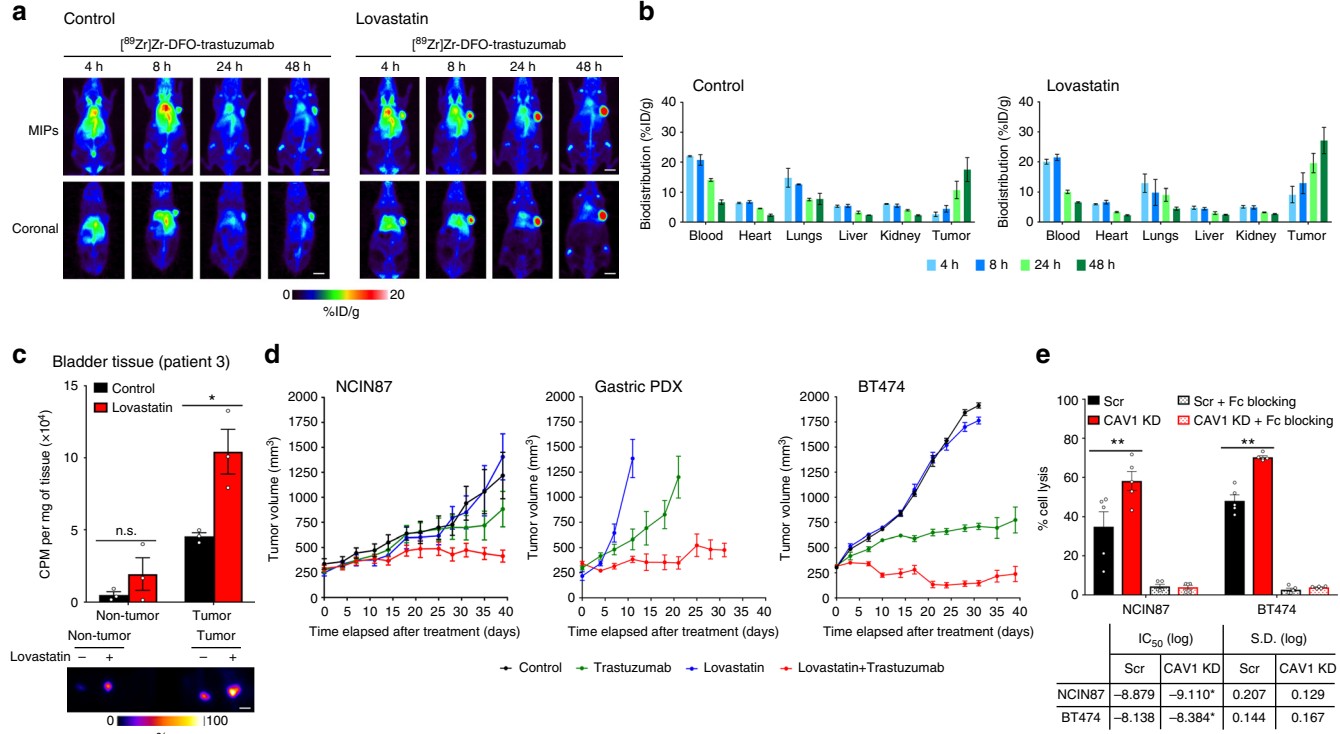

**Fig. 4** Statin treatment increases tumor uptake and efficacy of trastuzumab. **a** Representative coronal and MIPs PET images and **b** biodistribution data at 4, 8, 24, and 48 h p.i. of [$^{89}$Zr]Zr-DFO-trastuzumab in athymic nude mice bearing s.c. gastric tumors. Lovastatin (8.3 mg kg$^{-1}$ of mice) was orally administrated 12 h prior and at the same time as the tail vein injection of [$^{89}$Zr]Zr-DFO-trastuzumab (8.14–10.18 Mbq, 80–100 µg protein). Bars, $n = 4$ mice per group, mean ± S.E.M. MIPs maximum intensity projection. %ID g$^{-1}$, percentage of injected dose per gram. Scale bars: 5 mm. **c** Uptake (upper panel) and autoradiography (lower panel) of [$^{89}$Zr]Zr-DFO-trastuzumab in organotypic cultures of non-tumor and tumor human bladder tissues. Organotypic cultures were incubated with 25 µM of lovastatin for 4 h prior and at the same time as the addition of [$^{89}$Zr]Zr-DFO-trastuzumab. Slices were harvested at 3 h after incubation with [$^{89}$Zr]Zr-DFO-trastuzumab. Bars, $n = 3$, mean ± S.E.M, *$P < 0.05$ based on a Student's $t$-test. CPM counts per minute. Scale bars: 500 µm. **d** Superior in vivo therapeutic efficacy of trastuzumab combined with lovastatin when compared with trastuzumab in $nu/nu$ female mice bearing NCIN87 gastric and BT474 breast xenografts, and NSG mice bearing gastric PDXs ($n \geq 8$ mice per group). Intraperitoneal trastuzumab administration 5 mg kg$^{-1}$ weekly (during 5 weeks) was started at day 0. Lovastatin (4.15 mg kg$^{-1}$ of mice) was orally administrated 12 h prior and at the same time as the intraperitoneal injection of trastuzumab. **e** Ability of trastuzumab to mediate in vitro ADCC as determined with peripheral blood mononuclear cells (PBMCs) from healthy donors, in Scr and CAV1-depleted cancer cells (upper panel). Bars, $n = 5$, mean ± S.E.M, **$P < 0.01$ based on a Student's $t$-test. Trastuzumab IC$_{50}$ response as determined with engineered Jurkat cells stably expressing the FcγRIIIa receptor, in Scr and CAV1-depleted cancer cells (lower panel)

with lovastatin (Supplementary Fig. 7a, b). The tumor uptake of a radiolabeled isotype control IgG was significantly low and comparable in both CT mice as well as those treated with lovastatin (Supplementary Fig. 7c). These results indicate that a temporal depletion of CAV1 by lovastatin increases cell surface HER2, which in turn enhances the avidity of HER2-positive tumors for trastuzumab and accelerates tumor uptake of $^{89}$Zr-trastuzumab.

In addition to the conventional in vitro and in vivo studies, our findings were validated in organotypic cultures of fresh human bladder tumors. Tissues of HER2-positive bladder tumors were sliced immediately upon harvest during surgery, and grown in culture-plate inserts[41]. Viability assays of tumor slices revealed no significant loss of viable cells (Supplementary Fig. 8a, b) during 96 h of ex vivo culture ($P < 0.05$, Student's $t$-test). After ex vivo treatment with the active form of lovastatin, the tumor slices showed an increased uptake of $^{89}$Zr-trastuzumab ($P < 0.05$, Student's $t$-test; Fig. 4c). In non-tumor bladder tissue, lovastatin treatment did not significantly increase tissue accumulation of $^{89}$Zr-trastuzumab. Western blot analysis revealed a decrease in CAV1 protein expression after lovastatin treatment (Supplementary Fig. 8c), lending further support to our central hypothesis that temporal CAV1 depletion increases trastuzumab binding to

HER2-positive tumors. Additionally, through this experiment, we were able to demonstrate the utility of organotypic cultures as a potential ex vivo biological model to study the accumulation of target-specific radiolabeled antibodies in clinically relevant tumor tissues.

**CAV1 depletion enhances trastuzumab therapeutic efficacy.** To assess the efficacy of trastuzumab treatment when combined with CAV1 depletion in vivo, we conducted therapy studies in mice bearing NCIN87 gastric and BT474 breast, and gastric cancer PDXs. The gastric PDX model was established by implantation of HER2-positive and CAV1-positive gastric tumor tissue from Patient 1 (Fig. 1d) in immunodeficient NOD-SCID gamma (NSG) mice. Mice were treated weekly with 5 mg kg$^{-1}$ of trastuzumab for 5 weeks. Two doses of lovastatin (4.15 mg kg$^{-1}$) were orally administrated 12 h prior and at the same time as the intraperitoneal injection of trastuzumab. No difference was seen between the vehicle and lovastatin groups (Fig. 4d). Mice treated with either trastuzumab alone or trastuzumab combined with lovastatin showed tumor growth inhibition. However, the therapeutic efficacy of trastuzumab combined with lovastatin was significantly improved compared with trastuzumab alone. Taken

together, these results show that the in vivo therapeutic efficacy of trastuzumab is remarkably improved when the tumors are temporally depleted of CAV1—despite the latter being a transient effect.

One of the mechanisms of action of trastuzumab is by inducing tumor cell death via ADCC. A prerequisite to this phenomenon is the binding of trastuzumab to HER2 on tumor cells. Thus, trastuzumab-mediated ADCC is an effect that is closely related with HER2 density at cell membrane[42]. We attempted to study in vitro ADCC upon CAV1 depletion in NCIN87 gastric and BT474 breast cancer cells (T) using peripheral blood mononuclear cells (PBMCs) from healthy donors and engineered Jurkat cells as effector cells (E) that stably express the FcγRIIIa receptor. CAV1-depleted NCIN87 and BT474 cancer cells showed increased in vitro ADCC when PBMCs were used as effector cells (E:T ratio = 50:1, Fig. 4e). When engineered Jurkat cells were used as effector cells, CAV1 depletion in NCIN87 and BT474 cancer cells resulted in a decrease in IC$_{50}$ values (E/T = 15:1, Fig. 4e and Supplementary Fig. 9).

## Discussion

Endocytosis of receptors of the EGFR family is a complex biological process that remains to be completely understood. Nevertheless, intrinsic defects in this process have been demonstrated to influence therapeutic outcomes in EGFR-directed therapies[25,34,43]. Treatment of HER2-positive tumors with trastuzumab is premised on the presence of HER2 on the membrane of tumor cells. Therefore, the stability and availability of cell surface HER2 is an important predictor of trastuzumab efficacy. In the work at hand, we show that CAV1 knockdown reduces HER2 internalization and extends the half-life of HER2 at the cell membrane while concomitantly decreasing HER2 shedding without altering HER2 activity. Furthermore, accumulation of cell surface HER2 in response to pharmacological depletion of CAV1 was observed using preclinical xenograft models and organotypic cultures derived from freshly harvested human tumors. The discovery that CAV1 modulates cellular distribution of HER2 may have direct therapeutic implications for antibody-mediated targeting of HER2 and the use of such antibodies for molecular imaging and therapy of HER2-positive tumors.

CAV1 has been shown to regulate downstream signaling by members of the HER family[19,20,23,44] and interferes with the uptake/activity of a wide variety of antibodies or ADCs targeting the HER family[29,36,44–46]. Here, we show that the inverse correlation between cellular expression of CAV1 and HER2 directly affects the stability and localization of HER2 on the cell membrane. Our findings are supported by other reports that show CAV1 interaction with signaling molecules such as epidermal growth factor (EGF)[45,46]. Others have also previously demonstrated that although EGF is a ligand for its cognate receptor—EGFR, EGF treatment promotes the formation of heterodimers between EGFR and HER2, which activates HER2[47]. Further studies are necessary to evaluate the consequences of lovastatin treatment on the dimerization status of HER2.

Indeed, HER2 localization at the cell membrane is the product of a highly dynamic process[23] involving endocytosis, cytoplasmic recycling, and de novo synthesis. Furthermore, the membrane localization of HER2 is heterogeneous within tumors (Fig. 1b, c). Previous studies have reported that even at the single-cell level, individual HER2 molecules do not function in a uniform manner[48], and that this non-uniform behavior contributes to receptor heterogeneity. Equipped with this information, we analyzed patient tumor samples using immunofluorescence to understand how CAV1 expression patterns may relate to the cellular localization of HER2. Not surprisingly, a higher cell surface localization

of HER2 was found in human tumor tissues containing low levels of CAV1 (Fig. 1d). Conversely, tumors containing high levels of CAV1 had a low amount of HER2 localized at the cell membrane (Fig. 1d). Of course, a prerequisite for trastuzumab binding to HER2 is the latter's presence on the surface of a cancer cell and not within intracellular compartments. Taken together, these observations led us to hypothesize that CAV1 expression can be leveraged as a complementary biomarker to identify patients who would or would not benefit from HER2-targeted therapy with trastuzumab. Our findings in this study support previous reports that are suggestive of CAV1's role in the therapeutic response to HER2-targeted therapies[35,36,44,49,50]. Our studies clearly establish that downregulation of CAV1 increases HER2 availability at the cell surface.

Receptor-mediated endocytosis of ADCs such as TDM1 is central to the mechanism of action of this class of drugs. Indeed, the internalization of ADCs precedes the release of the cytotoxic drug within the lysosomal compartment of the cell, leading to cell death and manifestation of tumor regression. Others have demonstrated that the addition of nuclear localization sequence (NLS) peptides to trastuzumab labeled with an auger electron-emitter increases the internalization and nuclear localization of the antibody to decrease clonogenic survival of tumor cells[51] as well as to overcome insulin-like growth factor(IGF)-receptor-induced trastuzumab resistance in breast cancer cells[52]. In light of these reports, it would seem counter-productive to prevent HER2 internalization via pharmacological depletion of CAV1, as this might reduce the therapeutic benefit of HER2-directed ADCs. However, our data suggests that the temporal modulation of CAV1 extends the half-life of HER2 at the tumor cell surface to enhance the tumor's avidity for trastuzumab. The presence of such a target-rich sink could ultimately drive the in vivo pharmacokinetics of trastuzumab in favor of target-mediated drug disposition—leading to a significantly enhanced accumulation of trastuzumab in the tumor at early time points. Finally, being a transient and temporally controlled phenomenon that occurs under the influence of a pharmacological modulator of CAV1, normalization of the cellular distribution of HER2 including receptor-mediated internalization of the antibody bound to it would occur once the transient in vivo effect of the CAV1 modulator has worn out. Our finding that CAV1 depletion improves trastuzumab therapy, suggests that temporal modulation of CAV1 by a statin could be a strategy that can be applied in the clinic to improve trastuzumab binding/therapy for the treatment of HER2-positive tumors (Fig. 4). Admittedly, despite the successful in vitro demonstration of an effective trastuzumab-mediated ADCC potentiated by treatment with lovastatin, we realize that ADCC may not be a predominant mechanism of action contributing to the therapeutic efficacy of trastuzumab and/or combination of trastuzumab with lovastatin in our preclinical in vivo studies, since these experiments were performed using immunodeficient mice.

Statins, methyl-β-cyclodextrin, filipin, nystatin, and cholesterol oxidase are well known pharmacological inhibitors of caveolae/lipid-raft mediated endocytosis. Additionally, selective inhibition of lipid-raft/caveolae-mediated endocytosis can be achieved via acute depletion of the plasma membrane cholesterol. Depletion of cholesterol with methyl-β-cyclodextrin has been shown to release EGFR from lipid rafts, which increases the binding of iodine-125 ($^{125}$I)-labeled EGF[53]. Others have shown that cholesterol contributes to the cellular dynamics of HER2 and EGFR[33]. Here, we demonstrate that the bioavailability of HER2 at the cell membrane can be modulated via temporal depletion of cholesterol by lovastatin in ways that could serve to enhance trastuzumab binding and therapy against HER2-positive tumors (Figs. 3 and 4). For example, a reduction in the in vivo shedding of HER2

from proteolytic cleavage by sheddases might contribute to our observations of increased trastuzumab accumulation in HER2-positive tumors (Fig. 3c). HER2 shedding generates soluble truncated HER2 molecules in the serum[54]. [89]Zr-labeled trastuzumab has been proposed to bind the shed extracellular domain of HER2[55], which contains the binding site for trastuzumab. Notably, cholesterol depletion from cell membranes alters the function of enzymes with sheddase activity[56]. Taken together with our findings, it may be plausible that increased trastuzumab accumulation in membrane-bound fractions may also be a result of decreased shedding of HER2.

Despite being an effective modulator of CAV1 in our experiments, lovastatin is a lactone pro-drug that is plagued by poor oral bioavailability (< 5%) owing to its lipophilic nature[57]. After oral administration, lovastatin undergoes extensive first-pass metabolism and is hydrolyzed to the active beta-hydroxyacid form, which competitively inhibits HMGCR. HMGCR is the rate-limiting enzyme of cholesterol synthesis and is deregulated or elevated in several tumor types[58]. The bioavailability of lovastatin in systemic circulation is low and variable. It has a plasma half-life of 1.1–1.7 h, and takes approximately 2 to 4 h to reach maximum concentration in circulation. Only after metabolism and re-entry into systemic circulation, does lovastatin stand a chance to reach the tumor. The maximum human dose of lovastatin is 80 mg per day (1.33 mg kg$^{-1}$ per day for a 60 kg adult). The doses of lovastatin used in our imaging and therapy studies— 8.33 and 4.15 mg kg$^{-1}$—correspond to human equivalent doses of 0.68 and 0.34 mg kg$^{-1}$, which are significantly lower than the maximum recommended daily dose in humans. Previous preclinical studies have demonstrated the therapeutic potential of lovastatin per se using doses between 5 to 10 mg kg$^{-1}$ [59]. In our study, lovastatin was able to decrease CAV1 protein levels and increase cell membrane HER2 after 12 h of oral administration (Fig. 3). The low bioavailability of lovastatin and its short half-life suggests that a limited fraction of lovastatin would have reached the tumor after oral administration. It is expected that a more specific and efficient delivery of lovastatin to the tumors might further improve HER2 localization at the cell surface. Indeed, our findings support the surge in the interest for the use of statins for cancer treatment and the potential benefits thereof[41,58,60]. Since many cancer patients use statins, our data provides a rationale to pursue a clinical study combining lovastatin with trastuzumab to evaluate if this combination might positively affect the therapeutic outcomes or imaging strategies using trastuzumab in HER2-positive patients.

From the perspective of immunoPET, our study presents a major advance in the field of molecular imaging. Traditionally, the sluggish in vivo pharmacokinetics of full-length antibodies has necessitated their radiolabeling with long-lived positron-emitting radionuclides such as zirconium-89 ($t_{1/2} = 78.2$ h) to yield high contrast images between 72–120 h post-injection of the radioimmunoconjugate tracer. However, this manifests as a logistical inconvenience requiring patients to return to the clinic for a PET scan 4–5 days after they have been injected with the tracer[6]. However, we were able to obtain high contrast PET images as early as 4 h post-injection of [89]Zr-radiolabeled trastuzumab in HER2-positive tumors wherein CAV1 was pharmacologically modulated by using lovastatin (Fig. 4). Such an accelerated accumulation of [89]Zr-labeled trastuzumab is of tremendous clinical relevance from the standpoint of being able to overcome the current pharmacokinetic limitation of full-length antibodies to achieve same- to next-day immunoPET imaging in patients. This aspect further supports the rationale for combining a cholesterol-lowering drug such as a statin with trastuzumab to facilitate a relatively rapid turnaround time for the diagnosis of HER2-positive tumors via molecular imaging with full-length antibodies.

In conclusion, our study establishes the inverse relationship between CAV1 expression and HER2 localization at the cell membrane, and how this relationship can be pharmacologically modulated to augment HER2-targeted therapy of various cancers. Our findings suggest that CAV1 can be a promising biomarker to select patients for trastuzumab therapy. Finally, our preclinical data demonstrate the potential benefit of using statins to improve the therapeutic efficacy of trastuzumab and advance HER2-targeted molecular imaging in the clinic.

## Methods

**Cell lines and patient samples**. Gastric human cancer cell lines NCIN87 and KATOIII, breast cancer cell lines MDA-MB-231, SK-BR-3, MCF-7, and BT474 were purchased from American Type Culture Collection (ATCC). The human bladder cancer cell line, UMUC14, was obtained from Sigma-Aldrich. All cell lines were mycoplasma free and maintained at 37 °C in a humidified atmosphere at 5% $CO_2$. Cell lines utilized in this work were purchased in 2014–2016, and they were used within passage number of 15. All the cell lines were authenticated at Memorial Sloan-Kettering Cancer Center (MSKCC) integrated genomics operation core using short tandem repeat analysis. Deidentified patient gastric and bladder cancer samples were obtained from Memorial Hospital following IRB approval. PBMCs, obtained from the Immune Monitoring Facility at MSKCC, were separated from the blood of normal volunteers using lymphocyte separation medium (MP Biomedicals). Patients provided informed consent.

**Cell culture**. NCIN87 cells were grown in Roswell Park Memorial Institute (RPMI)-1640 growth medium supplemented with 10% fetal calf serum (FCS), 2 mM L-glutamine, 10 mM hydroxyethyl piperazineethanesulfonic acid (HEPES), 1 mM sodium pyruvate, 4.5 g L$^{-1}$ glucose and 1.5 g L$^{-1}$ sodium bicarbonate. KATOIII cells were grown in Iscove's Modified Dulbecco Medium (IMDM) growth medium supplemented with 20% FCS and 1.5 g L$^{-1}$ sodium bicarbonate. UMUC14 cells were grown in Minimum Essential Media (MEM) growth medium supplemented with 10% FCS, non-essential amino acids (NEAA), 2 mM L-glutamine, 1 mM sodium pyruvate, and 1.5 g L$^{-1}$ sodium bicarbonate. MDA-MB-231 cells were grown in Dulbecco's Modified Eagle's Medium-high glucose (DMEM-HG) supplemented with 10% FCS. SKBR3 cells were grown in McCoy's medium supplemented with 10% FCS, 1.5 mM glutamine, 2.2 g L$^{-1}$ sodium bicarbonate. MCF-7 cells were grown in Eagle's Minimum Essential Medium (EMEM) supplemented with 0.01 mg mL$^{-1}$ of human recombinant insulin and 10% FCS. BT474 cells were grown in a 1:1 mixture of DMEM:F-12 medium, supplemented with 10% FCS, 2 mM glutamine, NEAA. All cell culture media were supplemented with 100 units mL$^{-1}$ penicillin and streptomycin.

**Organotypic cultures of fresh human bladder tissues**. Fresh biopsy samples of bladder tissues were obtained immediately after surgical resection by the tissue procurement services at MSKCC. Normal and tumor bladder tissues were kept in organotypic cultures[41]. Tissue specimens were placed in Ham F-12 media supplemented with 20% inactivated FCS, 100 U mL$^{-1}$ penicillin, 100 μg mL$^{-1}$ streptomycin, 2.5 μg mL$^{-1}$ amphotericin B, and 100 μg mL$^{-1}$ of kanamycin. A Vibratome (Vibratome VT1200; Leica) was used to cut thin slices (300–500 μm) from biopsy specimens. Samples were soaked in ice-cold phosphate-buffered saline buffer (PBS) and immobilized using cyanoacrylate glue on the Vibratome platform. Tissue slices were cultured on organotypic inserts (Millipore) for up to 96 h at 37 °C in a 5% $CO_2$ humidified incubator. Five tissue slices were harvested at baseline time (T0) and thereafter, at 24 h intervals for viability assays using 1-(4, 5-dimethyltiazol-2-yl)-3, 5-diphenylformazan (MTT, Sigma-Aldrich) and CellTiter-Glo 3D (Promega) assays. For the MTT assay, tissue slices were incubated with 5 mg mL$^{-1}$ of MTT at 37 °C for 4 h, harvested, and the precipitated-salt extracted by incubation with 0.1 M hydrochloric acid (HCl)-isopropyl alcohol at room temperature for 25 min. A viability value was obtained by dividing the optical density of the formazan at 570 nm by the weight of the explants. Tissue viability was expressed as percentage of viability relative to that of samples at T0. For the CellTiter-Glo 3D cell viability assay, tissues were incubated with 100 μL of the CellTiter-Glo 3D reagent on a plate shaker for 5 min, followed by incubation for 30 min at room temperature. A viability value was obtained by dividing the luminescence signal by the weight of the explants.

**Transfection assays**. CAV1 was depleted using a pool of three target-specific 20–25 nt siRNA (Santa Cruz Biotechnology). Cancer cells were transfected in 6-well culture plates, at 60–80% confluence, with siRNA CAV1. Cells were also transfected with a scr siRNA in parallel as controls. For each transfection, cells were treated for 5 h with 2.4 μM of siRNA in transfection medium (Santa Cruz) containing 0.5 μL cm$^{-2}$ of transfection reagent (Santa Cruz).

CAV1 was amplified in cancer cells using CRISPR activation plasmid (h, Santa Cruz). Cancer cells were transfected in 6-well culture plates, at 60–80% confluence, with CRISPR Activation Plasmid CAV1. For each transfection, cells were treated for 5 h with 0.9 ng μL$^{-1}$ of CRISPR Activation Plasmid CAV1 in transfection medium (Santa Cruz) containing 0.5 μL cm$^{-2}$ of transfection reagent (Santa Cruz).

After incubation with siRNA CAV1 or CRISPR Activation Plasmid CAV1, complete media was added and the cells were incubated for 48 h. CAV1 downregulation or upregulation was evaluated 48 h post-transfection by western blotting.

Further experiments, to determine the effects of CAV1 depletion or amplification on HER2 stability at the cell membrane, were performed at 48 h post-transfection.

**Western blot analysis**. Whole-protein extracts from cells or tumors were prepared after cell scrapping or tissue homogenization, respectively, in radio-immunoprecipitation assay buffer [RIPA buffer: 150 mM sodium chloride (NaCl), 50 mM Tris hydrochloride (Tris-HCl), pH 7.5, 5 mM ethylene glycol tetraacetic acid (EGTA), 1% Triton X-100, 0.5% sodium deoxycholate (DOC), 0.1% sodium dodecyl sulfate (SDS), 2 mM phenylmethanesulfonyl (PMSF), 2 mM iodoacetamide (IAD), and 1 × protease inhibitor cocktail (Roche)]. After centrifugation at 16,000×g for 10 min at 4 °C, supernatants were used for protein quantification as determined with the Pierce BCA Protein Assay Kit (Thermo Fisher Scientific), followed by denaturation of the sample with Laemmli buffer. Following electrophoresis and transfer to polyvinylidene difluoride (PVDF) membranes (Bio-Rad), the blots were incubated in 5% (m/v) BSA in Tris-buffered saline buffer-Tween (TBS-T, Cell Signaling Technology) and probed with mouse anti β-actin 1:20,000 (Sigma, A1978), rabbit anti-HER2 1:800 (Abcam, ab131490), rabbit anti-HER2 phospho Y1139 1:500 (Abcam, ab53290), and rabbit anti-CAV1 1:500 (Abcam, ab2910) antibodies. After washing, the membranes were incubated with IRDye®800CW anti-Rabbit or anti-Mouse IgG 1:15,000 (LI-COR Biosciences) and imaged on the Odyssey Infrared Imaging System (LI-COR Biosciences) followed by densitometric analysis. Supplementary information contains uncropped scans of the blots shown in the figures.

**Biotin pull down of cell-surface proteins**. For biotin pull down assays, cells were washed twice with ice-cold PBS buffer containing 0.5 mM magnesium chloride (MgCl$_2$) and 1 mM calcium chloride (CaCl$_2$). Cells were incubated with 0.5 mg mL$^{-1}$ of EZ-LINK Sulfo-Biotin (Thermo Fisher Scientific) for 30 min at 4 °C with gentle rotation. The reaction was stopped by washing twice with 100 mM glycine (Thermo Fisher Scientific) in PBS containing 0.5 mM MgCl$_2$ and 1 mM CaCl$_2$. Cells were scrapped in RIPA buffer, lysates were centrifuged at 16,000×g for 10 min at 4 °C, and supernatants were collected and assayed for protein concentration using the Pierce BCA Protein Assay Kit. A volume of 500 μL of RIPA buffer containing equal amount of proteins was incubated with NeutrAvidin Agarose Resins (Thermo Fisher Scientific) for 2 h at 4 °C with gentle rotation and washed three times with RIPA buffer before suspension in Laemmli buffer.

**Isolation of caveolae-rich fractions**. Isolation of caveolae-rich fractions was done by discontinuous sucrose gradient (5−80% v/v) centrifugation using a caveolae/rafts isolation kit (Sigma-Aldrich). The gradient was centrifuged in a swing bucket rotor TH-641 (Sorvall) at 200,000×g for 4 h at 4 °C. Fractions were collected from the top to the bottom of the tube and analyzed by western blot.

**Endocytosis of biotinylated cell-surface proteins**. Following cell-surface biotinylation, endocytosis was initiated by the addition of pre-warmed cell culture medium in the presence of 1 μM trastuzumab at 37 °C for 15, 30, 60, and 90 min. Endocytosis was stopped by incubation of cells on ice for 10 min. For removal of non-internalized cell-surface biotin, cells were incubated with 50 mM Tris-HCl pH 8.7 [containing 20 mM dithiothreitol (DTT), 100 mM NaCl, 2.5 mM CaCl$_2$] for 20 min at 4 °C. Cells were rinsed with ice-cold PBS and scrapped in RIPA buffer. Biotinylated cell-surface HER2 was collected in NeutrAvidin Agarose Resins as described above.

**Protein–protein associations**. HER2 protein expression was correlated with CAV1 expression (using data from Cancer Cell Line Encyclopedia[37] or densitometric measurements of Western blot analysis) using a Spearman's rank correlation. Additionally, the STRING database (string-db.org) was used to determine HER2 protein – CAV1 protein interactions. High-confidence associations (score ≥ 0.700) were used to estimate the accuracy of the protein–protein networks.

**Immunofluorescence microscopy**. For immunocytochemistry, cells grown on coverslips were incubated with 1 μM trastuzumab for 90 min at 37 °C in the presence and absence of the active form of lovastatin. Cells were incubated with 25 μM of lovastatin for 4 h prior and at same time as the addition of trastuzumab. Cells were then permeabilized with 1% Triton X-100 in PBS (pH 7.4) and blocked with 5% bovine serum albumin in PBS buffer, before incubation with the DAPI and secondary fluorescent antibody goat anti-human IgG.

Immunofluorescence staining of HER2 and CAV1 were carried out on formalin-fixed, paraffin-embedded sections (10 μM) sections of human gastric tumors and NCIN87 s.c. tumors. Sections were submitted to MSKCC Molecular Cytology Core Facility for HER2, CAV1, hematoxylin and eosin staining.

**Cholesterol enrichment and caveolae depletion**. To enrich membrane cholesterol, cancer cells were exposed to methyl-β-cyclodextrin (MβCD, Santa Cruz Biotechnology) solution saturated with cholesterol[61] for 30 min before addition of trastuzumab. To inhibit caveolae-mediated pathways, cells were incubated with 2.5 mM MβCD or 0.5 μg mL$^{-1}$ filipin (Santa Cruz Biotechnology) for 30 min before addition of trastuzumab. For in vitro experiments with lovastatin, cells were incubated with 25 μM of the active form of lovastatin (Millipore) for 4 h prior addition of trastuzumab[40]. To determine whether mevalonic acid treatment rescued the lovastatin effect, the cell lines were treated with 25 μM lovastatin and 200 μM of R-Mevalonic Acid (Santa Cruz Biotechnology) for 4 h[62]. Cells were then treated with 1 μM [$^{89}$Zr]Zr-DFO-trastuzumab for 1.5 h.

**Conjugation and radiolabeling of trastuzumab**. Conjugation of trastuzumab with Indocyanine Green (ICG) was performed by incubation of trastuzumab with ICG at a molar ratio of 1:6 in 0.1 M Na$_2$HPO$_4$ (pH 8.6) at room temperature for 1.5 h, followed by purification with a size exclusion column (PD-10; GE Healthcare, Piscataway, NJ, USA). The concentration of ICG was calculated by measuring the absorption with the UV–Vis system to confirm the number of fluorophore molecules conjugated with each antibody molecule. The protein concentration was also determined by measuring the absorption at 280 nm with a UV–Vis system. BT474 CT and statin treated cells were incubated with trastuzumab—ICG conjugate and submitted to the Molecular Cytology Core Facility for live cell imaging.

Conjugation and radiolabeling of human IgG (Sigma-Aldrich) or trastuzumab with zirconium-89 ($^{89}$Zr) was achieved using the bifunctional chelate p-isothiocyanatobenzyl-desferrioxamine (DFO-Bz-NCS; Macrocyclics, Inc)[8]. Zirconium-89 was produced via proton beam bombardment of yttrium foil and isolated with high purity as $^{89}$Zr-oxalate at MSKCC[63].

**Internalization and saturation-binding assays**. For the internalization assays with [$^{89}$Zr]Zr-DFO-trastuzumab, cells were incubated with cell culture medium in the presence of 1 μM [$^{89}$Zr]Zr-DFO-trastuzumab for 90 min at 37 °C. Media containing non-cell-bound radiotracer was removed and the cells were washed twice with PBS. Cell surface-bound radiotracer was collected by cells incubation at 4 °C for 5 min in 0.2 M glycine buffer containing 0.15 M NaCl, 4 M urea at pH 2.5. Internalized fraction was obtained after cell lysis with 1 M sodium hydroxide (NaOH). Finally, the three fractions were measured for radioactivity on a gamma counter calibrated for $^{89}$Zr.

For the saturation-binding assays, cells were incubated with [$^{89}$Zr]Zr-DFO-trastuzumab (0–128 nM) in PBS (pH 7.5) containing 1% (m/v) human serum albumin (Sigma) and 1% (m/v) sodium azide (Acros Organics) for 3 h at 4 °C. Unbound radioactivity was removed and cells were washed three times with PBS. The cells were solubilized in 100 mM NaOH, recovered, and the total cell-bound radioactivity was measured on a gamma counter calibrated for $^{89}$Zr. Total binding was plotted vs. the concentration of [$^{89}$Zr]Zr-DFO-trastuzumab; the data were fit via non-linear regression with a one-site binding model in GraphPad Prism 7.00 to determine $B_{max}$, $K_D$ and the non-specific binding component. The non-specific component was subtracted from the total binding to generate specific binding curves (Fig. 2h).

**Uptake in organotypic cultures and autoradiography**. Slices of normal and tumor bladder tissues ($n = 4$) were harvested after 48 h of ex vivo culture and incubated with 25 μM of the active form of lovastatin for 4 h prior and at the same time as the addition of 1 μM [$^{89}$Zr]Zr-DFO-trastuzumab. Slices were harvested at 3 h after incubation with [$^{89}$Zr]Zr-DFO-trastuzumab, washed with PBS (pH 7.5) containing 0.1% (m/v) bovine serum albumin (BSA), air dried, weighed, and counted in a gamma counter calibrated for $^{89}$Zr. For autoradiography, slices were mounted in glycergel mounting medium, placed in a film cassette against phosphor imaging plate (Fujifilm BAS-MS2325; Fuji Photo Film) for 10 min at −20 °C. Phosphor imaging plates were read at a pixel resolution of 25 μm with a Typhoon 7000 IP plate reader (GE Healthcare).

**In vitro antibody-dependent cellular cytotoxicity (ADCC)**. Scr or CAV1-depleted NCIN87, UMUC14, BT474, and SK-BR-3 cancer cells ($1.7 × 10^4$) were preincubated with trastuzumab (1 μM) for 1.5 h at 37 °C in serum-free cell culture medium supplemented with 0.1% BSA before adding the effector cells in 1:1, 10:1, 25:1, or 50:1 effector (E) to target (T) ratios. PBMCs were used as effectors and cancer cells as targets. The cells were incubated for additional 48 h before determination of cell death using the Cytotoxicity Detection Kit (LDH; Roche). The percentage of cytotoxicity was calculated as follows: % cytotoxicity (experimental lysis−spontaneous effector lysis – spontaneous target lysis)/(maximum target lysis −spontaneous target lysis) × 100. Control Fc gamma receptor (FcyR) blocking studies were performed by pretreating PBMCs with 10 μg mL$^{-1}$ of human isotype IgG for 1 h before adding them to cancer cells in a ratio of 50:1.

Additional ADCC experiments were performed with the ADCC Reporter Bioassay, Core Kit (Promega), according to manufacture instructions. Briefly, Scr or CAV1 depleted NCIN87, UMUC14, BT474 and SK-BR-3 cancer cells were plated at the density of 5,000 cells/well in complete culture medium overnight before bioassay. On the day of the assay, the medium was removed, and then the series of concentrations of trastuzumab (0–10,000 ng mL$^{-1}$) were added to the cells, followed by addition of ADCC Bioassay Effector Cells (engineered Jurkat cells that stably express the FcγRIIIa receptor). The E:T ratio was 15:1, according to manufacture instructions. After 6 h of induction, luciferase assay reagent was added and luminescence determined using a luminometer. The data were fitted to a 4PL curve using GraphPad Prism software.

**Tumor xenografts.** All animals were treated according to the guidelines approved by the Research Animal Resource Center and Institutional Animal Care and Use Committee at Memorial Sloan Kettering Cancer Center, NY. Pereira PMR has a Category C accreditation for animal research given from Federation of European Laboratory Animal Science (FELASA). We adhere to the animal research: reporting of in vivo experiments (ARRIVE) guidelines and to the guidelines for the welfare and use of animals in cancer research. Eight- to 10-week-old *nu/nu* female mice (Charles River Laboratories) were injected subcutaneously with 2.5 million UMUC14, 5 million NCIN87 cells, or 5 million BT474 in a 150 μL cell suspension of a 1:1 (v/v) mixture of medium with reconstituted basement membrane (BD Matrigel, BD Biosciences). Drinking water of mice receiving estrogen receptor-positive cells BT474 was supplemented with 0.67 μg mL$^{-1}$ 17β-estradiol (Sigma) from 1 week in advance of inoculation until scarified. Fresh-estradiol supplemented water was provided twice a week. The mice were housed in type II polycarbonate cages, fed with sterilized standard laboratory diet and received sterile water ad libitum. The animals were housed at approximately 22 °C, 60% relative humidity, and a 12 h light, 12 h dark cycle was maintained. After arrival, all mice were allowed to acclimate to the facility's laboratory conditions for 1 week prior to experimentation.

Mice-bearing bilateral tumors were developed by subcutaneous injection of cancer cells on the bilateral dorsal flank regions using a sterile syringe with a 28-gauge needle. For experiments with mice-bearing unilateral tumors, cancer cells were subcutaneously injected on the right shoulder. The tumor volume ($V$/mm$^3$) was estimated by external vernier caliper measurements of the longest axis, $\alpha$/mm, and the axis perpendicular to the longest axis, $b$/mm. The tumors were assumed to be spheroidal and the volume was calculated in accordance with the equation $V = (4\pi/3) \times (\alpha/2)^2 \times (b/2)$.

An intratumoral injection of the active form of lovastatin (0.44 mg kg$^{-1}$ of mice) was performed in the pilot study with mice-bearing bilateral tumors. In further experiments with mice-bearing unilateral tumors, lovastatin was orally administered (8.3 mg kg$^{-1}$ of mice). The doses of lovastatin used in our study were lower when compared with the human maximum dose (80 mg per day): the 0.44 mg kg$^{-1}$ and 8.3 mg kg$^{-1}$ of mice doses correspond to daily human doses of 2.15 and 40.14 mg assuming an average human body weight of 60 kg. It should be noted that the intratumoral dose 0.44 mg kg$^{-1}$ of lovastatin was defined as 5% of oral dose 8.3 mg kg$^{-1}$ of mice, assuming 5% lovastatin bioavailability after oral administration.

When the volume of xenografts reached approximately 100 mm$^3$, mice were randomized into groups and treatments initiated. The active form of lovastatin (0.44 mg kg$^{-1}$ of mice) was intratumorally injected into the left-sided tumor of mice-bearing bilateral tumors (four mice per group) 4 h prior and at the same time as the tail vein injection of [$^{89}$Zr]Zr-DFO-trastuzumab (8.14–10.18 Mbq, 80–100 μg protein). The right-sided tumor was used as the CT and it was intratumorally injected with PBS.

For experiments in mice-bearing unilateral tumors, mice were randomly assigned into the following groups ($n = 4$ mice per group): Group I, oral administration of PBS 12 h prior to and at the same time as the tail vein injection of [$^{89}$Zr]Zr-DFO-trastuzumab (8.14–10.18 Mbq, 80–100 μg protein); Group II, oral administration of lovastatin (8.3 mg kg$^{-1}$ of mice) 12 h prior to and at the same time as the tail vein injection of [$^{89}$Zr]Zr-DFO-IgG (8.14–10.18 Mbq, 80–100 μg protein).

**Patient-derived xenograft (PDX) mouse models.** PDX models were established, by the Anti-tumor Assessment Core, from tumor specimens collected under an approved institutional review board protocol by the Research Animal Resource Center and Institutional Animal Care and Use Comittee at Memorial Sloan Kettering Cancer Center, NY. Briefly, tumors were minced, mixed with Matrigel, and implanted subcutaneously in 6- to 8-week-old female NSG mice (Jackson Laboratories). Once established, tumors were maintained and expanded by serial subcutaneous transplantation. Tumor samples were evaluated by immunohistochemistry and graded for HER2 and CAV1 expression.

**Acute biodistribution studies.** Acute biodistribution studies were performed after injection of radiolabeled trastuzumab. Mice were sacrificed and organs were harvested, weighed, and assayed in the gamma counter for biodistribution studies.

Radioactivity associated with each organ was expressed as percentage of injected dose per gram of organ (% ID g$^{-1}$)[8].

**Small-Animal positron emission tomography (PET) Imaging.** PET imaging experiments were conducted on a microPET Focus 120 scanner (Concorde Microsystems). Mice were anesthetized by inhalation of 1.5–2% isofluorane (Baxter Healthcare) in an oxygen gas mixture 10 min before recording PET images. PET data for each group ($n = 4$) was recorded, with mice under isofluorane anesthesia (1.5–2%), in list mode at 4, 8, 24, and 48 h after intravenous injection of [$^{89}$Zr]Zr-DFO-trastuzumab or [$^{89}$Zr]Zr-DFO-IgG. Images were analyzed using ASIPro VM software (Concorde Microsystems).

**In vivo therapeutic efficacy.** When tumor volumes reached 100 to 300 mm$^3$, mice were randomly grouped into treatment cohorts ($n \geq 8$ per group): vehicle, trastuzumab, lovastatin, or a combination of trastuzumab with lovastatin. Intraperitoneal trastuzumab administration 5 mg kg$^{-1}$ weekly (during 5 weeks) was started at day 0. Lovastatin (4.15 mg kg$^{-1}$ of mice) was orally administrated 12 h prior and at the same time as the intraperitoneal injection of trastuzumab. Tumor volumes were determined twice a week.

**Statistical analysis.** Data are expressed as mean ± S.E.M. Differences were analyzed by the Student t-test. The non-parametric one-tailed Spearman test was used to determine the correlation coefficient.

## Data availability

All data supporting the findings of this study are available in the Article, Supplementary Information, or upon request from the corresponding author. The Source Data underlying Figures in the main text and Supplementary Figures are provided as a Source Data file.

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

## Acknowledgements

We acknowledge the MSKCC Small-Animal Imaging Core Facility, the Radiochemistry and Molecular Imaging Probe Core, the Anti-tumor Assessment Core and Molecular Cytology Core Facility and the Immune Monitoring Facility, which were supported by NIH grant P30 CA08748. This study was supported in part by the Geoffrey Beene Cancer Research Center of MSKCC. We gratefully acknowledge Mr. William H. and Mrs. Alice Goodwin and the Commonwealth Foundation for Cancer Research and The Center for Experimental Therapeutics of Memorial Sloan Kettering Cancer Center. We thank Dr. Maurizio Scaltriti for reading the manuscript and providing insightful comments. We would also like to acknowledge Ricardo D'Oliveira Albanus from Department of Computational Medicine & Bioinformatics, University of Michigan for assistance in statistical analysis of the data. L.M.C. acknowledges support from the Ruth L. Kirschstein National Research Service Award postdoctoral fellowship (NIH F32-EB025050). P.M.R.P. and

S.K.S. gratefully acknowledge the Tow Foundation Postdoctoral Fellowships from the MSKCC Center for Molecular Imaging and Nanotechnology.

## Author contributions

Conception and design: P.M.R.P., J.S.L. Development of methodology: P.M.R.P., L.C., K.J.E., J.P., A.R. Acquisition of data (provided animals, acquired and managed patients, provided facilities, etc.): P.M.R.P., L.C., J.C.D, Y.Y.J., J.S.L. Analysis and interpretation of data (e.g., statistical analysis, biostatistics, computational analysis): P.M.R.P., L.C., S.K.S., J.S.L. Writing, review, and/or revision of the manuscript: P.M.R.P., L.C., K.J.E., S.K.S., J.C.D., Y.Y.J., J.S.L. Study supervision: J.S.L.

## Additional information

**Competing interests:** The authors declare no competing interests.

