## [Peer Review File · Nature Communications]

Reviewers' Comments:

Reviewer #1:

Remarks to the Author:

Summary:

The current paper investigates the relationship between HER2 and caviolin-1 (CAV1). The premise is that response to trastuzumab therapy may be governed in part by differential rates of endocytosis and recycling, and thereby the available membrane associated pool. The authors find that CAV1 expression is inversely correlated with HER2 protein level various cell lines – information acquired from the Cancer Cell Line Encyclopedia and generated in-house. This was confirmed by IF of tissue from two different human gastric tumors. Interestingly, trastuzumab treatment increased the interaction between HER2 and CAV1. When CAV1 is reduced by siRNA, HER2 was slightly increased, but when CAV1 was overexpressed, HER2 expression was significantly reduced. Interestingly, the half-life of membrane-bound HER2 was increased when CAV1 was reduced. An inhibitor of HMGCR, lovastatin, was used to reduce membrane cholesterol content and thus membrane CAV1. Acute, intermittent treatment led to a general increase in HER2 localization to the membrane. Intriguingly, lovastatin pretreatment of mice bearing HER2+ tumors greatly increased imaging intensity of ⁸⁹Zr-labeled trastuzumab. Similar results were found when PET was used to image [18F]AIF-NOTA-PEG-trastuzumab. Finally, lovastatin administration increased the efficacy of trastuzumab in terms of decreased growth of HER2+ tumors in mice.

Overall, this paper describes an interesting relationship between CAV1 and HER2. The data support the authors' conclusions. Striking results are presented from experiments where lovastatin was used to indirectly decrease membrane bound CAV1 (via reduced cholesterol), thereby increasing HER2 at the membrane. Strong data is presented indicating the use of lovastatin to increase the tumoral imaging with labelled trastuzumab, and increase the efficacy of this therapy in reducing tumor growth. Thus, the authors demonstrate the potential therapeutic utility of altering CAV1 to enhance the effects of anti-HER2 therapy.

Major Concerns:

- 1) Although the paper does a nice job at demonstrating that HER2 localization is altered when CAV1 is, the authors do not demonstrate whether HER2 activity is changed. Does the increase in membrane HER2 after CAV1 loss result in altered HER2 activity (increased or decreased)? Altered interaction with HER3 etc? Etc. This would be very important information prior to therapeutic translation.
- 2) The dose of lovastatin used both in vitro and in vivo needs to be justified. The authors use a dose of 25uM in vitro, while the IC50 of lovastatin for HMGCR is ~3-4nM. The chosen in vivo dose is also very high. As the authors point out, most statins are heavily metabolized by first pass metabolism, making their circulating concentrations very low. Doses in humans are restricted due to significant side effects. Thus, it is unclear whether the effects the authors observe with high dose statins would be achievable in human patients.
- 3) In vitro lovastatin treatment is very high. Should the effects be on target, they should be 'rescued' by mevalonate treatment.
- 4) Alternate cholesterol depletion methods (eg: cyclodextrin) should be used to confirm the lovastatin findings. Furthermore, the authors' findings would be strengthened by testing whether increasing membrane cholesterol content has the opposite effects.
- 5) Although the images are of high quality, some of the described changes are subtle. Quantification and statistical analysis should be performed (eg: Figs. 2, 4).

Reviewer #2:

Remarks to the Author:

Comments for Authors

The manuscript by Pereira et al. describes the interaction of caveolin-1 with HER2 and a putative role that it may have in determining the effectiveness of trastuzumab therapy. The novelty of the findings only marginally extends our current understanding of the interaction between caveolin-1 and HER2. Indeed, published work from Chao's group (references 37 and 42) where T-DM1 was utilized as a therapeutic agent predicts most of the results of the current study. Briefly, these investigators demonstrated that increasing caveolin-1 expression increases sensitivity to T-DM1 and they elegantly show that the converse is also true when caveolin-1 expression is silenced. The data presented in the manuscript by Pereira et al. is not nearly as compelling. Moreover, it is not surprising that decreasing caveolin-1 expression renders HER2 over expressing cell lines more sensitive to trastuzumab-mediated ADCC. Pereira et al. attempt to make this point with data shown in Fig 5H. It is noted that a statistically significant difference in ADCC is only shown at one E:T ratio, with one cell line, at one fixed concentration of trastuzumab. The xenograft (5I) study with lovastatin and trastuzumab treatment is a critical experiment for the current manuscript. The combination effect of lovastatin and trastuzumab treatment is only marginally better than trastuzumab treatment alone. Again, data from only one xenograft model is described.

The authors misinterpret or selectively cite the literature. This is most apparent as it pertains to the effectiveness and mechanisms of action of trastuzumab. For example, in the Discussion, where the authors state "... the therapy is only effective in a low percentage of patients...." exactly what are they referring to? In the adjuvant setting, recent trastuzumab trials have shown that the 3 year DFS is greater than 90% and OS is greater than 95%. In first line metastatic breast cancer median PFS is greater than 18 months and OS is greater than 56 months. There is always room for improvement but statements such as the one above suggests that the authors are unaware of the progress that has been made over the last 20 years in the treatment of HER2-positive breast cancer.

The authors' lack of awareness of the basic scientific literature is comparable to their ignorance of clinical literature. Trastuzumab is widely acknowledged to have multiple mechanisms of action that include ADCC, anti-signaling properties and the inhibition of HER2 shedding. Indeed an interesting area of investigation would be to determine the role of CAV1 expression in HER2 ligand-independent and ligand-dependent signaling.

In summary, this manuscript does not represent the quality of work that is generally published in Nature Communications and may be more suitable, after extensive revision, for a specialty journal. Selected specific criticisms are listed below. Neither the authors nor the editors should consider this a comprehensive list but a rather a sampling of areas that require improvement.

Selected Specific Comments:

1. Fig 1A- it is curious that the investigators did not include more breast cancer and gastric cancer cell lines in their correlation of CAV1 with HER2 expression. In particular, breast cancer cell lines with a wide range of HER2 expression such as MCF7, MDA-231, ZR75, MDA-175, MDA 453, MDA 361, SK-BR-3, BT-474, KPL4 etc. should be included to determine if the inverse correlation of CAV1 and HER2 expression holds up. Similarly, it is peculiar that the gastric cell lines KATO III and MKN7 are also not included. The data should also be included in Supplementary Fig 1A.
2. Fig 1F- The method used for immunoprecipitation of HER2 is a problem. Trastuzumab is used both as a treatment agent and as a primary antibody for immunoprecipitation. The following data need to be included: 1) the entire western blot for the HER2 ip, not just the well cropped 185 kDa band; 2) a control lane where trastuzumab is omitted as an ip antibody and without trastuzumab treatment but the sample is still processed over protein G. I predict that the authors will be

surprised by the results. Bottom line: immunoprecipitation experiments should be run with an antibody that does not compete with trastuzumab and ideally reacts with the C-terminus of HER2 AND where the antibody is covalently bound to a resin.

3. Fig. 1I- How many replicates were performed in this experiment. How many times was the entire experiment performed? At least two more high expressing cell lines should be included in the half-life analysis.

4. Fig. 2A- The western blot for the CAV1 detection with caveolin-1 KD is suspicious. It looks like transfer from the gel to the PVDF membrane was incomplete. Half of the band that was transferred looks to be of similar intensity to the band without KD.

5. Fig. 2B- Looks like the CAV1 band in the control lane has disappeared. How does one rectify this result with that shown in Figs. 1F and 2A?

6. Fig. 2C & 2D- Additional time points (e.g. 36h and 60h) are required to definitively determine whether CAV-1 knockdown increases HER2 half-life.

7. Fig. 3- The authors misinterpret their data. When CAV1 expression is silenced the Kd for trastuzumab binding to HER2 is 3.0 nM. When CAV1 is present the Kd is 1.2 nM. Therefore, CAV1 knockdown decreases (not increases) the affinity of trastuzumab for HER2. Notwithstanding this error, how does the cytoplasmic interaction of CAV1 alter the affinity of an extracellular binding antibody to HER2? I suspect that there is no difference in trastuzumab binding \pm CAV1 expression and if the authors repeated the binding experiments a number of times then there would not be a statistically meaningful difference in Kd.

8. Fig 5I- The combination effect of lovastatin and trastuzumab is unimpressive. The experiment should be repeated with at least two additional tumor models. Also, the X-axis should be plotted with equal spacing between numbers (sometimes its 4 and sometimes its 3).

Reviewer #3:

Remarks to the Author:

Summary

This paper describes the relationship between caveolin-1 (CAV-1) expression and HER2 internalization in bladder (UMIC14) and gastric cancer (NCIN87) cell lines and tumors in mice. It is hypothesized that CAV-1 is responsible for HER2 internalization, and that blocking CAV-1 using siRNA or lovastatin, a cholesterol synthesis inhibitor, will decrease HER2 internalization and increase the levels of HER2 on the cell surface. The paper employs a number of techniques to assess the effect of inhibiting CAV-1 on HER2 expression including immunofluorescence for CAV-1 and HER2, Western blot for HER2, as well as radioligand binding assays using ⁸⁹Zr-trastuzumab which binds to HER2. Tumor imaging studies using PET with ⁸⁹Zr-trastuzumab as well as biodistribution studies are also used to assess the effects of lovastatin on increasing HER2 in UMIC14 and NCIN87 tumor xenografts in mice. The immunofluorescence and Western blot data is not that convincing for increasing the membrane-associated HER2, but the PET imaging does appear to show improved imaging of tumors with ⁸⁹Zr-trastuzumab in the presence of lovastatin while increasing HER2 expression by about 2-fold in the two cancer cell lines and in tumors in vivo. However, there appears to be lower background radioactivity in the blood pool, liver and other organs on images obtained with lovastatin treatment which improves the image quality while the tumor intensity looks similar compared to control mice not treated with lovastatin. These differences are not evident on the biodistribution data, which results in a discordance with the PET imaging. There is also a therapy study which shows increased effectiveness of trastuzumab against HER2-positive tumor xenografts, possibly explained by increased HER2 expression in the presence of lovastatin. Finally, there is a study of pre-targeted imaging using a trastuzumab analogue that binds to a subsequently administered ¹⁸F-labeled small molecule by click chemistry. The pre-targeting approach showed successful tumor imaging but the tumor uptake (2-4% ID/g) and tumor:blood ratios (less than 1) are not ideal for imaging by PET.

In summary, the paper presents an interesting study in which it is demonstrated that HER2

expression may be modulated by inhibiting CAV-1 or by administration of lovastatin, and that increased HER2 improves tumor imaging and treatment of HER2-positive tumors in mice. The lovastatin intervention to increase HER2 is particularly interesting as it is a clinically used drug that could improve tumor uptake of ^{89}Zr -trastuzumab for PET and unlabeled trastuzumab for cancer treatment.

From a presentation point-of-view, I feel that there are far too many figures with too many details of the different experiments. There needs to be a more concise and clear presentation of the studies performed to test the hypothesis. In addition, I feel that the pre-targeting approach should not be included in this paper, as it further complicates the description of studies to test the main hypothesis. These pre-targeting studies should be reported in a separate paper.

Major Points

- P. 5. It is stated that the mechanism of promotion of HER2 internalization by trastuzumab involves trastuzumab-mediated activation of the receptor and phosphorylation. I don't think this is correct. To my knowledge, trastuzumab inhibits the activation of HER2 caused by homo- or heterodimerization initiated by activation of the EGFR or other Type 1 growth factor receptors. The authors provide reference 34 for this statement, but this paper does not appear to evaluate trastuzumab, but rather other HER2 antibodies.
- Fig. 1A. I am not sure that this figure should be in the main manuscript, since it is not an experimental result. It is a comparison between CAV-1 and HER2 expression on different cell lines from the Cancer Cell Line Encyclopedia. It should be moved to the Supplemental Figures. Also, I don't understand why the top panel is on a Log₂ scale and the bottom one on a Log₁₀ scale.
- Fig. 1B. It is stated that there is heterogeneous (cell membrane and intracellular HER2) but the figure is not that clear, and it is in B&W. Why? The figure needs to be improved and magnified.
- Fig. 1C. I am not convinced that the figure shows an inverse relationship between CAV1 and HER2 spatially in the tumors. There are some areas that are both CAV1-positive and HER2-positive. This comparison could be technically effected by the intensity of the fluorescence signal for CAV1 and HER2 in different regions of the tumor, unrelated to differences in CAV-1 or HER2.
- Fig. 1D and E. I am not convinced that there is a difference in HER2 expression in the two patient tumors – one from a CAV-1-high tumor and one from a CAV-1-low tumor. I agree that there is a difference in CAV-1 expression between the two tumors.
- Fig. 1F – I do not understand the significance of this figure.
- Fig. 2A – This figure is supposed to show increased HER2 with siRNA knock-down of CAV-1, but I don't see how it shows this – HER2 seems similar with/without knock-down, although CAV-1 is lower.
- Fig. 2G – The percent internalized ^{89}Zr -trastuzumab in the absence of CAV-1 knock-down seems very high (80%). In my experience, the internalization of radiolabeled trastuzumab is about 30-50%. Do NC1N87 cells internalize trastuzumab with unusual efficiency?
- Fig. 3A – This binding assay is not showing saturation, and thus the fitting of the data to a 1-site binding model to determine B_{max} may not be accurate. Supplementary Fig. 2H is much more convincing for an increase in the B_{max} value for HER2 binding with CAV-1 knock-down.
- Fig. 3B, C, F. These are LigandTracer binding plots. This technology is not widely used and the authors provide no interpretation of the plots, thus I would recommend deleting all of the LigandTracer plots in the paper. A saturation radioligand binding assay is the "gold standard" for evaluating K_D and B_{max} and is more readily appreciated by scientists in the field.
- Supplementary Fig. 3 should be deleted. There is no explanation of these LigandTracer binding curves to interpret the data, and as mentioned above, this is a non-standard method to evaluate binding affinity.
- Fig. 3D (Table). K_D values should be given to 1-decimal place. Also, there appears to be a major discrepancy between the B_{max} values which are " $\times 10^4$ " and the HER2s per cell which are " $\times 10^5$ ". Is there an error here? Same for Supplementary Table 1 – there may be an error in the B_{max} units.
- Fig. 4A. The images of the cells need to be magnified to appreciate differences in internalized

HER2 in control vs. lovastatin treated cells.

- Fig. 4E, F. These figures should be in color. Also, what is the significance of the two separate sections? Are they two sections from the same tumor xenograft, or sections from two different xenografts? Please state in the legend.
- Fig. 5A. It seems that the background radioactivity in the liver, blood pool and other normal organs is lower for the lovastatin-treated mice, while the tumor intensity is similar for control and lovastatin-treated mice. Improved imaging with lovastatin appears to be due mainly to lower background.
- Supplementary Fig. 4. Tumor intensity looks the same on the lovastatin-treated and control tumors in the mice on the coronal and transaxial sections. It is not clear that the lovastatin intratumoral injection increased the tumor uptake of ⁸⁹Zr-trastuzumab (at least on the images). Panels C and D are the tumor and normal tissue uptake, which do show a difference in tumor uptake with lovastatin treatment.
- Supplementary Fig. 5. This shows the tumor and normal tissue biodistribution for control and lovastatin-treated mice. This is a key figure and should be in the main manuscript. There appears to be a modest increase in tumor uptake of ⁸⁹Zr-trastuzumab, but only slight differences in normal tissue uptake. However, this doesn't seem to agree with the images shown in Fig. 5A and Supplementary Fig. 5A, which show lower background in mice treated with lovastatin. The authors should explain this apparent discrepancy.
- Fig. 5G. I am not enthusiastic about including the pre-targeting approach in this paper, as it is a separate study that requires controls to demonstrate specificity as well as optimization of the sequence of the trastuzumab and ¹⁸F-labeled small molecule administration etc.
- P. 11. Pre-targeting results. As mentioned above, I am not that enthusiastic to include the pre-targeting experiments in this paper.
- Supplementary Fig. 7. There is tumor imaging with the pre-targeting approach, but increased tumor uptake in the presence of lovastatin is still low (4% vs. 2% ID/g). Tumor: blood ratios are only slightly greater than 1 with lovastatin and much less than 1 without lovastatin. These are actually very low for a pre-targeting approach, which is normally intended to yield very high tumor: blood ratios. Also, at this early imaging time point (<4 h), it is not clear what proportion of tumor localization is due to specific binding to trastuzumab pre-bound to HER2 in the tumor, or due to tumor blood flow.
- Fig. 5I. This study does not show an ADCC effect as the mice are immunocompromised. It likely shows an increased action of trastuzumab by other mechanisms such as inhibition of tumor growth signaling. There needs to be more details in the legend of this figure to indicate the type of mice, tumor xenograft, dose administration etc.
- P. 12. The authors need to point out that the in vivo effect of trastuzumab in mice with/without lovastatin is not due to ADCC since immunocompromised mice are used, and also the human Fc-domain of trastuzumab may not activate ADCC in mice.
- P. 14. The authors should point out that interfering with HER2 internalization may decrease the effectiveness of trastuzumab-emtansine, a new drug used for HER2-positive breast cancer.
- P. 14. The statement that HER2 internalization is not required for the therapeutic activity of trastuzumab is not correct, since one proposed mechanism of action of the drug is promotion of HER2 internalization, thereby decreasing HER2 on the cell surface, and thus decreasing HER2-mediated growth signaling.
- P. 14. The statement that ADCC is the major mechanism of action of trastuzumab is not correct. There are many different mechanisms of action proposed for trastuzumab, including inhibition of growth signalling, promotion of HER2 internalization, decreased angiogenesis, stabilization of the ECD of HER2 avoiding a truncated constitutively activated receptor etc.
- P. 15. The statement that it takes a few days to weeks until images with good contrast are obtained with radiolabeled antibodies is absolutely incorrect. Good quality images can be obtained as early as 48 h after administration of radiolabeled intact IgG forms (sometimes as early as 24 h). Radiolabeled antibody fragments (Fab or F(ab')₂) clear from the blood and normal tissues quickly, and good quality images can be obtained as early as 6 h and certainly within 12-24 h.
- P. 16. As mentioned previously, I would not include the pre-targeting approach.

- P. 24. I think the dose of lovastatin is higher than the normal human dose. The human dose is 10-80 mg/day which corresponds to 0.14-1.4 mg/kg/day based on a 70 kg human, whereas the authors administered 4.15 mg/kg to mice.

Minor Points

- P. 4. References 7,14 pertain to PET imaging with ⁸⁹Zr-trastuzumab and do not include imaging with ⁸⁹Zr-trastuzumab emtansine.
- P. 4. Statement that "antibodies are cell membrane impermeable" is not quite accurate, since there is receptor-mediated internalization for some antibodies.
- P.4. Statement that current testing for HER2 evaluates overall HER2 expression (cell surface and intracellular) is not correct. The criteria for HER2 scoring relates to the intensity and proportion of cells with cell surface immunohistochemical staining or in situ hybridization techniques (e.g. FISH) to assess HER2 gene amplification.
- P. 5 line 135. "de" should be "the".
- Fig. 1-I. The half-lives should be given in whole numbers or maximum one decimal place.
- Fig. 2C –The effect of CAV-1 siRNA knock-down only slightly increases HER2 on the cell membrane (i.e. in the presence of CHX to inhibit protein synthesis).
- Fig. 3D – This figure is not needed, and should just be explained in the text.
- Fig 4C is not needed. Please just explain in the methods section of the text.
- Fig. 5F is not needed. Please just explain in the methods section of the text.
- P. 14. Some references for trastuzumab-mediated radioimmunotherapy are needed. Also, there are examples of where interfering with HER2 internalization could decrease the effectiveness of radioimmunotherapy, for example some papers that have been published using low energy Auger electron emitters, such as ¹¹¹In.

Reviewer #1: Cholesterol metabolism

Summary: *The current paper investigates the relationship between HER2 and caviolin-1 (CAV1). The premise is that response to trastuzumab therapy may be governed in part by differential rates of endocytosis and recycling, and thereby the available membrane associated pool. The authors find that CAV1 expression is inversely correlated with HER2 protein level various cell lines – information acquired from the Cancer Cell Line Encyclopedia and generated in-house. This was confirmed by IF of tissue from two different human gastric tumors. Interestingly, trastuzumab treatment increased the interaction between HER2 and CAV1. When CAV1 is reduced by siRNA, HER2 was slightly increased, but when CAV1 was overexpressed, HER2 expression was significantly reduced. Interestingly, the half-life of membrane-bound HER2 was increased when CAV1 was reduced. An inhibitor of HMGCR, lovastatin, was used to reduce membrane cholesterol content and thus membrane CAV1. Acute, intermittent treatment led to a general increase in HER2 localization to the membrane. Intriguingly, lovastatin pretreatment of mice bearing HER2+ tumors greatly increased imaging intensity of 89Zr-labeled trastuzumab. Similar results were found when PET was used to image [18F]AIF-NOTA-PEG-trastuzumab. Finally, lovastatin administration increased the efficacy of trastuzumab in terms of decreased growth of HER2+ tumors in mice.*

Overall, this paper describes an interesting relationship between CAV1 and HER2. The data support the authors' conclusions. Striking results are presented from experiments where lovastatin was used to indirectly decrease membrane bound CAV1 (via reduced cholesterol), thereby increasing HER2 at the membrane. Strong data is presented indicating the use of lovastatin to increase the tumoral imaging with labelled trastuzumab, and increase the efficacy of this therapy in reducing tumor growth. Thus, the authors demonstrate the potential therapeutic utility of altering CAV1 to enhance the effects of anti-HER2 therapy.

Response: We are very grateful to the reviewer for the positive overview of the manuscript and constructive suggestions to improve the paper.

Major Concerns:

1) *Although the paper does a nice job at demonstrating that HER2 localization is altered when CAV1 is, the authors do not demonstrate whether HER2 activity is changed. Does the increase in membrane HER2 after CAV1 loss result in altered HER2 activity (increased or decreased)? Altered interaction with HER3 etc? Etc. This would be very important information prior to therapeutic translation.*

Response: We thank the reviewer for these insightful comments. Taking these comments into consideration, we have now included new data regarding HER2 activity after the cells have been treated with lovastatin. Please see Fig. 2I and pages 9, 13:

Page 9: “In our studies, the active form of lovastatin effectively depleted CAV1 expression in NCIN87 cells without altering phosphorylated HER2 levels (Fig. 2I).”

Page 13: “In the work at hand, we show that CAV1 knockdown reduces HER2 internalization, resulting in a substantial accumulation of HER2 at the cell membrane, a concomitant decrease in HER2 shedding and improved ADCC in vitro without altering HER2 activity.”

We are currently investigating if lovastatin treatment alters HER2 interaction with other members of the HER family. This work is extensive given the need for optimization of the experimental protocols, the number of HER antigens, as well as the number of HER-targeting antibodies (e.g., pertuzumab) that we want to include in this work. Although this would be an extremely interesting aspect to investigate, we believe that it does not fall within the scope of the current manuscript and warrants publication as a separate report at a later date.

2) *The dose of lovastatin used both in vitro and in vivo needs to be justified. The authors use a dose of 25uM in vitro, while the IC50 of lovastatin for HMGCR is ~3-4nm. The chosen in vivo dose is also very high. As the authors point out, most statins are heavily metabolized by first pass metabolism, making*

their circulating concentrations very low. Doses in humans are restricted due to significant side effects. Thus, it is unclear whether the effects the authors observe with high dose statins would be achievable in human patients.

Response: Although the IC₅₀ of lovastatin for HMGCR is ~3-4nM, several *in vitro* studies aiming to deplete cholesterol have reported the use of lovastatin in the μM range (please see the following references: Svelto et al. *Am J Physiol Renal Physiol* 298: F266–F278, 2010; Huttner et al. *Traffic* 2000 1: 952–962; Chattopadhyay et al. *Scientific Reports* volume 7, Article number: 4484 2017).

The *in vitro* dose of lovastatin used in our experiments was based on previous reports that demonstrated 25 μM as the pharmacologic dose capable of depleting caveolin-1 protein. Kindly see page 9 and reference 40:

Page 9: “Previous studies have reported that 25 μM of lovastatin reduces the protein expression of CAV1 *in vitro* [40].”

To determine the *in vivo* dose of lovastatin, the following rationale was taken into account: Assuming that 5% lovastatin is bioavailable after oral administration, an intratumoral dose of 0.44 mg/kg of mice was deduced as 5% of an oral dose – i.e. 8.3 mg/kg of mice. Kindly see page 24:

Page 24: “The doses of lovastatin used in our study were lower when compared with the human maximum dose (80 mg/day): the 0.44 mg/kg and 8.3 mg/kg of mice doses correspond to daily human doses of 2.15 and 40.14 mg assuming an average human body weight of 60 kg. It should be noted that the intratumoral dose 0.44 mg/kg of mice was defined as 5% of oral dose 8.3 mg/kg of mice, assuming 5% lovastatin bioavailability after oral administration.”

The maximum human dose of lovastatin is 80 mg/day, which corresponds to 1.33 mg/kg/day based on a 60 kg body weight. For conversion of the human dose to a mouse dose as *per FDA guidelines*, this dose is adjusted for differences in metabolic rates between man and mouse based on body surface area. Kindly see Nair and Jacob, *Journal of Basic and Clinical Pharmacy*, 2016, 7. The doses of lovastatin used in both imaging and therapeutic studies - 8.33 mg/kg and 4.15 mg/kg correspond to human equivalent doses of 0.68 and 0.34 mg/kg. Both of these are remarkably lower than the maximum human dose. *Notably, patients use lovastatin on a daily basis, while in our study lovastatin was administrated twice a week.*

Also, previous preclinical studies have demonstrated the therapeutic potential of lovastatin *per se* using small animal doses between 5 to 10 mg/kg.

Changes have been made to the text to ensure clarity and avoid misunderstandings. Kindly see page 15:

Page 15: “The maximum human dose of lovastatin is 80 mg/day (1.33 mg/kg/day for a 60 kg adult). The doses of lovastatin used in our imaging and therapy studies – 8.33 and 4.15 mg/kg – correspond to human equivalent doses of 0.68 and 0.34 mg/kg, which are significantly lower than the maximum recommended daily dose in humans. Previous preclinical studies have demonstrated the therapeutic potential of lovastatin *per se* using doses between 5 to 10 mg/kg [56].”

3) In vitro lovastatin treatment is very high. Should the effects be on target, they should be ‘rescued’ by mevalonate treatment.

Response: Taking this very interesting and valid comment into consideration, we have included new data using ⁸⁹Zr-labeled trastuzumab to bind HER2 on NCIN87 gastric, UMUC14 bladder, BT474 breast and SKBR3 breast cancer cells treated with lovastatin in the presence of mevalonate. Kindly, see Figs. 3A, Supplementary Fig. 4D, page 9:

Page 9: “Lovastatin’s inhibition of the enzyme 3-hydroxyl-3-methylglutarylcoenzyme A (HMG-CoA) reductase (HMGCR) is known to result in the decrease of mevalonate and its downstream products (e.g. cholesterol). In the presence of lovastatin, cells exhibited an increase in membrane-associated ⁸⁹Zr-labeled trastuzumab, an effect that is rescued by the addition of mevalonate to the cell culture (Fig. 3A; Supplementary Fig. 4D).”

4) *Alternate cholesterol depletion methods (eg: cyclodextrin) should be used to confirm the lovastatin findings. Furthermore, the authors’ findings would be strengthened by testing whether increasing membrane cholesterol content has the opposite effects.*

Response: These are excellent suggestions and we have now added new data regarding ⁸⁹Zr-labeled trastuzumab binding to NCIN87 gastric, UMUC14 bladder, BT474 breast and SKBR3 breast cancer cells treated with cyclodextrin, filipin and cholesterol. Kindly, see Figs. 2G, 3A Supplementary Figs. 4B, 4D, pages 8,9:

Page 8: “Increased membrane-associated ⁸⁹Zr-labeled trastuzumab was also observed upon depletion of cholesterol using methyl-β-cyclodextrin (MβCD) and filipin (Fig. 2G; Supplementary Fig. 4B).”

Page 9: “In the presence of lovastatin, cells exhibited an increase in membrane-associated ⁸⁹Zr-labeled trastuzumab, an effect that is rescued by the addition of mevalonate to the cell culture (Fig. 3A; Supplementary Fig. 4D). Additional studies demonstrated that an increment in membrane cholesterol content has the opposite effect (Fig. 3A; Supplementary Fig. 4D).”

5) *Although the images are of high quality, some of the described changes are subtle. Quantification and statistical analysis should be performed (eg: Figs. 2, 4).*

Response: Western Blot and Immunofluorescence assays were used to yield *qualitative* readouts in our study. We do agree that the changes in HER2 expression (as determined by cell surface biotinylation assays) at the cell membrane are subtle. However, our *quantitative* methods, including new data from additional cell lines, show that CAV1 depletion does increase the half-life of HER2 at the cell surface, resulting in increased membrane-associated HER2 molecules *per cell* and subsequent trastuzumab binding. Please, see page 7:

Page 7” Cell-surface biotinylation experiments were performed to analyze HER2 expression and localization in the plasma membrane. siRNA-mediated knockdown of CAV1 slightly increased HER2 at the cell membrane for NCIN87 cells (Fig. 2A) and UMUC14 cells (Supplementary Fig. 3B), without alteration of total HER2 protein levels. In contrast, when we used clustered regularly interspaced short palindromic repeats (CRISPR) activation plasmid to increase CAV1, we found that forced CAV1 overexpression promotes loss of HER2 at the cell membrane (Fig. 2B; Supplementary Fig. 3C). Whilst CAV1 depletion resulted in a slight increase in cell-surface HER2, a more pronounced effect of knocking down CAV1 was demonstrated by the extended half-life ($P < 0.05$) of HER2 at the cell membrane (Fig. 2C, 2D; Supplementary Fig. 3D, 3E) in NCIN87, UMUC14 and BT474 cancer cells.”

Reviewer #2: Herceptin

Comments for Authors

The manuscript by Pereira et al. describes the interaction of caveolin-1 with HER2 and a putative role that it may have in determining the effectiveness of trastuzumab therapy. The novelty of the findings only marginally extends our current understanding of the interaction between caveolin-1 and HER2. Indeed, published work from Chao’s group (references 37 and 42) where T-DM1 was utilized as a therapeutic agent predicts most of the results of the current study. Briefly, these investigators demonstrated that

increasing caveolin-1 expression increases sensitivity to T-DM1 and they elegantly show that the converse is also true when caveolin-1 expression is silenced.

Response: The paper by Chao et al, PlosOne, 2015 uses HER2-positive BT474 and SKBR3 breast cancer cells to determine whether or not caveolin-1 (CAV1) protein interferes with uptake and toxicity of trastuzumab emtansine (T-DM1). Intriguing data is reported in this manuscript since the authors report that SKBR3 cells have a high expression of CAV1. In our present study, CAV1 knockdown did not change HER2 half-life at the SKOV3 cell membrane because this cell line has few caveolae structures. Kindly see references Brown et al., Mol Biol Cell. 2010, 21, 2226–2240; van Deurs et al., Mol Biol Cell. 2004, 15,1557–1567, all of which corroborate our own data shown in the following figure:

Furthermore, the article by Chao et al (Scientific Reports, 2018) used conventional *in vitro* approaches to determine if CAV1 interferes with T-DM1 internalization. In this report, they show that uptake of T-DM1 was increased in BT474 cells pretreated with metformin as a result of upregulated CAV1 expression.

From left to right: MCF7, MDAMB231, KATOIII, BT474, SKBR3

However, neither paper specifically investigated the role of CAV1 in HER2 cell-membrane availability/stability, nor did the authors further explore/validate their preliminary *in vitro* findings using preclinical *in vivo* models. We believe that HER2 dynamics and heterogeneity are complex and cannot be sufficiently mimicked using conventional *in vitro* experimental platforms. Thus, we contend that our work bears significant novelty and extends the translational potential of the preliminary findings beyond those reported by Chao et al.

The data presented in the manuscript by Pereira et al. is not nearly as compelling. Moreover, it is not surprising that decreasing caveolin-1 expression renders HER2 over expressing cell lines more sensitive to trastuzumab-mediated ADCC. Pereira et al. attempt to make this point with data shown in Fig 5H. It is noted that a statistically significant difference in ADCC is only shown at one E:T ratio, with one cell line, at one fixed concentration of trastuzumab. The xenograft (5I) study with lovastatin and trastuzumab treatment is a critical experiment for the current manuscript. The combination effect of lovastatin and trastuzumab treatment is only marginally better than trastuzumab treatment alone. Again, data from only one xenograft model is described.

Response: We most respectfully disagree with the reviewer's opening statement of this critique. We believe that our work clearly presents compelling experimental evidence to establish the strong inverse relationship between CAV1 and HER2. Furthermore, we demonstrate how the molecular dynamics of this relationship can be pharmacologically modulated to tailor and/or enhance HER2-targeted molecular imaging and therapy.

To the reviewer's second half of this critique, we would like to clarify that the observation of a statistically significant difference in ADCC previously shown at an E/T ratio of 50/1 alone was a limitation of our experimental set up. This may be attributed to the fact that the use of PBMCs can often be prone to inherent donor-to-donor variability. Previous reports in the literature have demonstrated that the ADCC response might be different between fresh, frozen/overnight rested

and frozen/overnight not rested PBMCs. Please see reference Baum et al., J Immunol Methods. 2014, 406:1-9. In our studies, PBMCs were obtained from unidentified donors and we do not have information about their processing from the time of collection leading up to the moment that we obtained them from the core facility at MSKCC for our experiments.

In the revised version of the manuscript we provide new data to demonstrate ADCC with PBMCs at a ratio of 50/1 (E:T) using NCIN87, BT474, SKBR3 and UMUC14 cancer cells. Additionally, in an attempt to determine IC₅₀ values while circumventing the experimental variability arising from PBMCs obtained from healthy donors, we have performed additional ADCC experiments using engineered Jurkat cells that stably express the FcγRIIIa receptor. Kindly see Figs. 4E, Supplementary Fig. 8, Page 12:

Page 12: “We therefore attempted to study in vitro ADCC upon CAV1 depletion in NCIN87 gastric and BT474 breast cancer cells (T) using peripheral blood mononuclear cells (PBMCs) from healthy donors and engineered Jurkat cells as effector cells (E) that stably express the FcγRIIIa receptor. CAV1-depleted NCIN87 and BT474 cancer cells showed increased in vitro ADCC when PBMCs were used as effector cells (E:T ratio = 50:1, P = 0.0032, Fig. 4E). When engineered Jurkat cells were used as effector cells, CAV1 depletion in NCIN87 and BT474 cancer cells resulted in a decrease in IC₅₀ values (E/F = 15:1, Fig. 4E; Supplementary Fig. 8).”

Two additional preclinical models were included in our in vivo therapeutic studies. Please see Fig. 4D, Page 12:

Page 12: “To assess the efficacy of trastuzumab treatment when combined with CAV1 depletion in vivo, we conducted therapy studies in mice bearing HER2-positive and CAV1-positive tumors (NCIN87 gastric and BT474 breast, and gastric cancer PDXs). The gastric PDX model was established by implantation of HER2-positive and CAV1-positive gastric tumor tissue from Patient 1 (Fig. 1D) in immunodeficient NOD-SCID gamma mice.”

The authors misinterpret or selectively cite the literature. This is most apparent as it pertains to the effectiveness and mechanisms of action of trastuzumab. For example, in the Discussion, where the authors state “... the therapy is only effective in a low percentage of patients...” exactly what are they referring to? In the adjuvant setting, recent trastuzumab trials have shown that the 3 year DFS is greater than 90% and OS is greater than 95%. In first line metastatic breast cancer median PFS is greater than 18 months and OS is greater than 56 months. There is always room for improvement but statements such as the one above suggests that the authors are unaware of the progress that has been made over the last 20 years in the treatment of HER2-positive breast cancer.

Response: We most sincerely regret the misunderstanding. The manuscript has been carefully reviewed to ensure clarity and avoid any misunderstanding or lack of our appreciation for the clinical benefit to patients from the use of pre-existing as well as evolving HER2-targeted therapies to treat breast cancer. We do describe trastuzumab as a very effective therapy. Kindly see page 4:

Page 4: “Prior to the development of targeted anti-HER2 therapy, patients with HER2-positive tumors demonstrated reduced disease-free survival compared to patients whose tumors expressed low levels of HER2 [3]. These findings established HER2 as a therapeutic target and a tumor biomarker. Over the past two decades clinical evidence has unequivocally demonstrated that the inhibition of this oncogene improves treatment outcomes, and has led to the emergence of several effective anti-HER2 therapies [4].”

The authors' lack of awareness of the basic scientific literature is comparable to their ignorance of clinical literature. Trastuzumab is widely acknowledged to have multiple mechanisms of action that include ADCC, anti-signaling properties and the inhibition of HER2 shedding. Indeed an interesting area of investigation would be to determine the role of CAV1 expression in HER2 ligand-independent and ligand-dependent signaling.

Response: We regret the oversight. The manuscript has been carefully reviewed to ensure clarity and avoid misunderstandings. Kindly, see pages 4:

Page 4: “Clinically, the antitumor activity of trastuzumab is attributed to more than a single mechanism of action. Direct action of the antibody drug is premised on receptor downregulation and subsequent alterations to intracellular signaling including attenuation of downstream pro-tumorigenic cell signaling, inhibition of HER2 shedding, and inhibition of tumor angiogenesis. On the other hand, indirect action due to activation of an immune response via antibody dependent cell-mediated cytotoxicity (ADCC) has also been proposed as a mechanism of action for this drug [15-17].”

Further to citing the mechanisms of action implicated in the therapeutic efficacy of trastuzumab, we investigated the role of ADCC as a mechanism of action leading to improved response to therapy when trastuzumab was combined with lovastatin. This was considered a plausible mechanism because trastuzumab-induced ADCC effect is highly dependent on the presence of HER2-bound antibody at the cell membrane and its availability for interaction with immune effector cells in the tumor microenvironment.

Another possible mechanism is now included in our revised manuscript where lovastatin treatment demonstrates reduced HER2 shedding. Kindly, see Fig. 3C, page 9:

Page 9: “Since the extracellular domain of HER2 is known to shed in vivo, we evaluated the impact of lovastatin treatment on this phenomenon. Specifically, treatment with lovastatin reduced HER2 shedding by approximately 50% in vitro using NCIN87 cells versus 70% in NCIN87 xenografts (Fig. 3C).”

The suggestion of the reviewer about further experiments to determine the role of CAV1 expression in HER2 ligand-independent and ligand-dependent signaling is valid, but is not the scope of the present manuscript. We would respectfully suggest that investigation of this aspect belongs to another paper.

In summary, this manuscript does not represent the quality of work that is generally published in Nature Communications and may be more suitable, after extensive revision, for a specialty journal. Selected specific criticisms are listed below. Neither the authors nor the editors should consider this a comprehensive list but a rather a sampling of areas that require improvement.

Response: We sincerely believe and hope that the reviewer will find the revised form of our manuscript suitable for publication in Nature Communications. To merit that status, we have made an earnest attempt to address the reviewer’s comments and have extensively revised the manuscript. Please read further.

Selected Specific Comments:

1. Fig 1A- it is curious that the investigators did not include more breast cancer and gastric cancer cell lines in their correlation of CAV1 with HER2 expression. In particular, breast cancer cell lines with a wide range of HER2 expression such as MCF7, MDA-231, ZR75, MDA-175, MDA 453, MDA 361, SK-BR-3, BT-474, KPL4 etc. should be included to determine if the inverse correlation of CAV1 and HER2 expression holds up. Similarly, it is peculiar that the gastric cell lines KATO III and MKN7 are also not included. The data should also be included in Supplementary Fig 1A.

Response: We have now included additional cancer cell lines to demonstrate the inverse correlation between CAV1 and HER2. Unfortunately, some of the cell lines suggested by the reviewer are not available in our laboratory at this point in time. Nevertheless, those cell lines are included in the spearman correlation plot based on information obtained from a reliable public database – the cancer cell line encyclopedia (CCLE). Kindly, see Figs. 1A, B and Supplementary Fig. 1A.

2. *Fig 1F- The method used for immunoprecipitation of HER2 is a problem. Trastuzumab is used both as a treatment agent and as a primary antibody for immunoprecipitation. The following data need to be included: 1) the entire western blot for the HER2 ip, not just the well cropped 185 kDa band; 2) a control lane where trastuzumab is omitted as an ip antibody and without trastuzumab treatment but the sample is still processed over protein G. I predict that the authors will be surprised by the results. Bottom line: immunoprecipitation experiments should be run with an antibody that does not compete with trastuzumab and ideally reacts with the C-terminus of HER2 AND where the antibody is covalently bound to a resin.*

Response: Considering the comments from reviewers #2 and #3, we feel that the setup of our immunoprecipitation studies may have its limitations, which prevent it from adding value to our manuscript. Thus, this data has been omitted to avoid any misunderstandings.

3. *Fig. 1I- How many replicates were performed in this experiment. How many times was the entire experiment performed? At least two more high expressing cell lines should be included in the half-life analysis.*

Response: We agree with reviewer's suggestion that additional HER2-positive cancer cell lines need to be included in our analysis of the half-life of HER2 at the cell membrane. Per the reviewer's suggestion, we carried out additional experiments to determine the half-life of HER2 in HER2-positive BT474 and SKBR3 cancer cells (n = 3 independent experiments). Kindly, see Supplementary Fig. 2.

4. *Fig. 2A- The western blot for the CAV1 detection with caveolin-1 KD is suspicious. It looks like transfer from the gel to the PVDF membrane was incomplete. Half of the band that was transferred looks to be of similar intensity to the band without KD.*

Response: We agree with reviewer's comment that our previous image may have been suggestive of incomplete transfer of protein from the gel to the membrane. A more representative image is now included in the revised version of the manuscript. Kindly see Fig. 2A.

5. *Fig. 2B- Looks like the CAV1 band in the control lane has disappeared. How does one rectify this result with that shown in Figs. 1F and 2A?*

Response: We reckon that it is not fair to perform inter-blot comparisons. We would also like to clarify that the total amount of protein loaded for the gel shown in Fig. 2B was significantly lesser than what was used for blots shown in Fig. 2A and Fig.1G. This change was made due to the fact that the signal obtained after CAV1 amplification was at the saturation limit compared to the total amount of protein from the sample containing endogenous levels of CAV1.

6. *Fig. 2C & 2D- Additional time points (e.g. 36h and 60h) are required to definitively determine whether CAV-1 knockdown increases HER2 half-life.*

Response: The two additional time points suggested by the reviewer are now included in our analysis. Kindly see Figs. 1G, 2C, Supplementary Figs. 2,3.

7. *Fig. 3- The authors misinterpret their data. When CAV1 expression is silenced the Kd for trastuzumab binding to HER2 is 3.0 nM. When CAV1 is present the Kd is 1.2 nM. Therefore, CAV1 knockdown decreases (not increases) the affinity of trastuzumab for HER2. Notwithstanding this error, how does the cytoplasmic interaction of CAV1 alter the affinity of an extracellular binding antibody to HER2? I suspect that there is no difference in trastuzumab binding \pm CAV1 expression and if the authors repeated the binding experiments a number of times then there would not be a statistically meaningful difference in Kd.*

Response: The reviewer makes a great point here. We have updated this section to clarify the role of CAV1 depletion on trastuzumab binding. The K_D values are indeed, as the reviewer suggested, not significantly different when control cells are

compared with CAV1-depleted cells. However, CAV1 depletion with siRNA and lovastatin *significantly* increased B_{max} (i.e., number of HER2 molecules per cell), which in this analysis would render the tumor cells more avid for trastuzumab binding. Kindly see page 8:

Page 8: “Next, the binding profile of ^{89}Zr -labeled trastuzumab in competitive radioligand saturation binding assays confirmed that CAV1 knockdown increased HER2 density on the cell membrane (B_{max}) (Fig. 2H; Supplementary Fig. 4C). Specifically, ^{89}Zr -labeled trastuzumab binding to NCIN87 cells revealed a 1.4-fold more HER2 in siRNA-treated CAV1 NCIN87 cells and 1.8-fold higher HER2 in UMUC14 cells versus their scr-treated counterparts. Together, these data indicate that CAV1 depletion stabilizes HER2 at the cell membrane and improves the binding avidity for trastuzumab on HER2-positive cancer cells.”

8. Fig 5I- The combination effect of lovastatin and trastuzumab is unimpressive. The experiment should be repeated with at least two additional tumor models. Also, the X-axis should be plotted with equal spacing between numbers (sometimes its 4 and sometimes its 3).

Response: Following the reviewer’s suggestion, we performed additional therapeutic studies in BT474 xenografts and gastric PDXs. These two models are both HER2-positive and CAV1-positive. Our previous scale on the X-axis was with the days at which the tumors were measured and the axis is now corrected with equal spacing between numbers, as per the reviewer’s suggestion. Kindly see Fig. 4D, page 11:

Page 11: “To assess the efficacy of trastuzumab treatment when combined with CAV1 depletion in vivo, we conducted therapy studies in mice bearing HER2-positive and CAV1-positive tumors (NCIN87 gastric and BT474 breast, and gastric cancer PDXs). The gastric PDX model was established by implantation of HER2-positive and CAV1-positive gastric tumor tissue from Patient 1 (Fig. 1D) in immunodeficient NOD-SCID gamma mice.”

Reviewer #3: Imaging

Summary

This paper describes the relationship between caveolin-1 (CAV-1) expression and HER2 internalization in bladder (UMIC14) and gastric cancer (NCIN87) cell lines and tumors in mice. It is hypothesized that CAV-1 is responsible for HER2 internalization, and that blocking CAV-1 using siRNA or lovastatin, a cholesterol synthesis inhibitor, will decrease HER2 internalization and increase the levels of HER2 on the cell surface. The paper employs a number of techniques to assess the effect of inhibiting CAV-1 on HER2 expression including immunofluorescence for CAV-1 and HER2, Western blot for HER2, as well as radioligand binding assays using ^{89}Zr -trastuzumab which binds to HER2. Tumor imaging studies using PET with ^{89}Zr -trastuzumab as well as biodistribution studies are also used to assess the effects of lovastatin on increasing HER2 in UMIC14 and NCIN87 tumor xenografts in mice.

The immunofluorescence and Western blot data is not that convincing for increasing the membrane-associated HER2, but the PET imaging does appear to show improved imaging of tumors with ^{89}Zr -trastuzumab in the presence of lovastatin while increasing HER2 expression by about 2-fold in the two cancer cell lines and in tumors in vivo.

However, there appears to be lower background radioactivity in the blood pool, liver and other organs on images obtained with lovastatin treatment which improves the image quality while the tumor intensity looks similar compared to control mice not treated with lovastatin. These differences are not evident on the biodistribution data, which results in a discordance with the PET imaging.

There is also a therapy study which shows increased effectiveness of trastuzumab against HER2-positive tumor xenografts, possibly explained by increased HER2 expression in the presence of lovastatin. Finally, there is a study of pre-targeted imaging using a trastuzumab analogue that binds to a subsequently administered ^{18}F -labeled small molecule by click chemistry. The pre-targeting approach showed

successful tumor imaging but the tumor uptake (2-4% ID/g) and tumor: blood ratios (less than 1) are not ideal for imaging by PET.

In summary, the paper presents an interesting study in which it is demonstrated that HER2 expression may be modulated by inhibiting CAV-1 or by administration of lovastatin, and that increased HER2 improves tumor imaging and treatment of HER2-positive tumors in mice. The lovastatin intervention to increase HER2 is particularly interesting as it is a clinically used drug that could improve tumor uptake of ⁸⁹Zr-trastuzumab for PET and unlabeled trastuzumab for cancer treatment.

From a presentation point-of-view, I feel that there are far too many figures with too many details of the different experiments. There needs to be a more concise and clear presentation of the studies performed to test the hypothesis. In addition, I feel that the pre-targeting approach should not be included in this paper, as it further complicates the description of studies to test the main hypothesis. These pre-targeting studies should be reported in a separate paper.

Response: We are very grateful to the reviewer for the positive overview of the manuscript and constructive suggestions to improve the paper.

Major Points

• *P. 5. It is stated that the mechanism of promotion of HER2 internalization by trastuzumab involves trastuzumab-mediated activation of the receptor and phosphorylation. I don't think this is correct. To my knowledge, trastuzumab inhibits the activation of HER2 caused by homo- or heterodimerization initiated by activation of the EGFR or other Type 1 growth factor receptors. The authors provide reference 34 for this statement, but this paper does not appear to evaluate trastuzumab, but rather other HER2 antibodies.*

Response: We thank the reviewer for noticing this and we have now corrected the introduction in the manuscript. Please, see page 5:

Page 5: "Notably, cell-surface receptors involved in tumor development are characterized by abnormal trafficking from the cell membrane to intracellular compartments [19, 20]. Distinct from HER2, endocytosis of the other members of the HER family occurs after ligand binding [20]. Although HER2 has no known ligand, the "open" conformation of the extracellular domain contributes to the dynamics of the HER2 surface pool [21, 22]. The localization of HER2 at the cell membrane is a heterogeneous and dynamic process [19, 23, 24] governed by differential rates of endocytosis and recycling [20, 24, 25]. In addition to cell membrane expression, HER2 localizes in the cytoplasm [26] and nucleus [27]."

• *Fig. 1A. I am not sure that this figure should be in the main manuscript, since it is not an experimental result. It is a comparison between CAV-1 and HER2 expression on different cell lines from the Cancer Cell Line Encyclopedia. It should be moved to the Supplemental Figures. Also, I don't understand why the top panel is on a Log2 scale and the bottom one on a Log10 scale.*

Response: Fig. 1A showing data from CCLE was moved to the supplementary information file and it is now Supplementary Fig. 1A. The X-axis of Fig. 1A and Supplementary Fig. 1A has been adjusted to a log2 scale, as per the reviewer's suggestion.

• *Fig. 1B. It is stated that there is heterogeneous (cell membrane and intracellular HER2) but the figure is not that clear, and it is in B&W. Why? The figure needs to be improved and magnified.*

Response: Since only one channel (green) was being visualized in this image we felt it would be best represented on a grayscale. However, per the reviewer's suggestion we have magnified two sections of this image, which are now rendered green. We have modified the manuscript to avoid misunderstandings. Please see page 6:

Page 6: We observed that HER2 does not exhibit predominant membrane localization in all cell clusters of NCIN87 (HER2-positive gastric cancer cell line [8]) tumor xenografts (Fig. 1B).

• *Fig. 1C. I am not convinced that the figure shows an inverse relationship between CAV1 and HER2 spatially in the tumors. There are some areas that are both CAV1-positive and HER2-positive. This comparison could be technically effected by the intensity of the fluorescence signal for CAV1 and HER2 in different regions of the tumor, unrelated to differences in CAV-1 or HER2.*

Response: Fig. 1C shows that in CAV1-positive and HER2-positive cells, HER2 localization at the cell membrane is not predominant (highlighted by the dashed white circles numbered 2 and 4). Conversely, HER2 staining in the membrane is pronounced in cells that have low levels of CAV1 indicated by less red signal in the immunofluorescence image (highlighted by the dashed white circles numbered 1 and 3). In addition to demonstrating an inverse relationship between CAV1 and HER2, this figure aims to exemplify the heterogeneity in expression of these two inversely correlated proteins at the cell membrane and shows that HER2 is not predominant at the cell membrane in cells where CAV1 is present.

We contend that a technical defect is unlikely to have occurred, since the seesaw effect between CAV1 and HER2 was observed in cells that are present within the same cell cluster as well – for example, cluster #4). We have modified the wording in the manuscript to avoid misunderstandings. Please see page 6:

Page 6: “In cancer cells where CAV1 is absent, HER2 is exclusively present at the cell membrane (Fig. 1C highlighted by the dashed white circles 1 and 3). On the other hand, cancer cells expressing CAV1 exhibit reduced HER2 staining at the cell membrane (Fig. 1C highlighted by the dashed white circles 2 and 4).”

• *Fig. 1D and E. I am not convinced that there is a difference in HER2 expression in the two patient tumors – one from a CAV-1-high tumor and one from a CAV-1-low tumor. I agree that there is a difference in CAV-1 expression between the two tumors.*

Response: The reviewer raises a good point. While the difference in CAV-1 expression between the two patient tumor samples is obvious from the red fluorescence signal, it can be hard to say the same for HER2 staining represented by the distribution of the green fluorescence signal in the 2 tumors. Thus, we had included the overlay to provide a fairly good view of the distinct staining of the tissues from both these fluorescence channels as well as DAPI (blue) to indicate presence of nucleated cells in the sections being analyzed and presented.

As a matter of fact, we had 6 such patient tumor samples that remarkably affirmed the case in point – the predominance of green versus red staining of tumor tissue arising from HER2 versus CAV-1 expression was inversely correlated. Patient samples having low CAV-1 (red fluorescence) consistently yielded high HER2 staining (green fluorescence) and vice versa.

To better tell the difference in the green fluorescence between the tumors from patient 1 versus patient 2, we quantified the fluorescent intensity of the entire tumor from both these samples. Quantification of fluorescent signal in the entire tumor section using Panoramic Flash 250 slide scanner from 3DHitech (Budapest, Hungary) - Zeiss 20x/0.8NA objective, and further analysis by the Molecular Cytology Core Facility at MSKCC, demonstrated that the expression of HER2 was significantly lower in patient #1 versus patient #2. The result of this quantitative analysis of the immunofluorescence images has now been included as Fig. 1D.

Furthermore, HER2 staining appears to occur predominantly at the cell membrane of the tumor sample from patient #2 since CAV1 expression (indicated by red fluorescence) is low in this tumor. High magnification (60x) images of the cells have now been included as insets in the fluorescence images to show predominant HER2 localization at the surface of cancer cells in the tumor section derived from patient #2 versus diffused and internalized cytoplasmic staining of cancer cells in the tumor section derived from patient #1.

Most interestingly, upon conducting therapy studies using the tumor tissue from patient #1 to generate gastric PDX models we were able to demonstrate that inhibition of CAV1 with lovastatin significantly increases its response to treatment with trastuzumab. Kindly see Fig. 4D, pages 6/7:

Pages 6/7: “Consistent with these findings, immunofluorescence staining of HER2-positive tumor samples obtained from patients with gastric cancer revealed that cells with high expression of CAV1 have low membrane staining of HER2 (Fig. 1D). In HER2-positive tumor samples having low expression levels of CAV1, HER2 exhibits predominant membrane staining. Together, the immunofluorescence analyses of preclinical and clinical samples indicate that CAV1 expression plays a role in HER2 antigen density at the target cell membrane.”

- *Fig. 1F – I do not understand the significance of this figure.*

Response: Considering the comments made by reviewers #2 and #3, we have omitted the results of the immunoprecipitation studies, as they do not seem to add additional value to our manuscript.

- *Fig. 2A – This figure is supposed to show increased HER2 with siRNA knock-down of CAV-1, but I don't see how it shows this – HER2 seems similar with/without knock-down, although CAV-1 is lower.*

Response: A similar comment was made by reviewer #1.

We agree with both reviewers that the changes in HER2 expression at the cell membrane (as determined by cell surface biotinylation assays) are subtle. However, using *quantitative* methods, including new data from additional cell lines, we were able to show that CAV1 depletion does increase the half-life of HER2 at the cell membrane, yielding higher amount of membrane-associated HER2s *per cell* to result in increased trastuzumab binding to such cells. Please see page 7:

Page 7: “siRNA-mediated knockdown of CAV1 slightly increased HER2 at the cell membrane for NCIN87 cells (Fig. 2A) and UMUC14 cells (Supplementary Fig. 3B), without alteration of total HER2 protein levels. In contrast, when we used clustered regularly interspaced short palindromic repeats (CRISPR) activation plasmid to increase CAV1, we found that forced CAV1 overexpression promotes loss of HER2 at the cell membrane (Fig. 2B; Supplementary Fig. 3C). Whilst CAV1 depletion resulted in a slight increase in cell-surface HER2, a more pronounced effect of knocking down CAV1 was demonstrated by the extended half-life ($P < 0.05$) of HER2 at the cell membrane (Fig. 2C, 2D; Supplementary Fig. 3D, 3E) in NCIN87, UMUC14 and BT474 cancer cells.”

- *Fig. 2G – The percent internalized ^{89}Zr -trastuzumab in the absence of CAV-1 knock-down seems very high (80%). In my experience, the internalization of radiolabeled trastuzumab is about 30-50%. Do NCIN87 cells internalize trastuzumab with unusual efficiency?*

Response: Great observation and comment. We attribute the high degree of internalization of trastuzumab to be a result of high CAV1 expression in NCIN87 cells as well as the concentration (1 μM) of ^{89}Zr -labeled trastuzumab used in this experiment. At lower concentrations, our internalization assays agree fairly well with the reviewer's experience.

- *Fig. 3A – This binding assay is not showing saturation, and thus the fitting of the data to a 1-site binding model to determine B_{max} may not be accurate. Supplementary Fig. 2H is much more convincing for an increase in the B_{max} value for HER2 binding with CAV-1 knock-down.*

Response: Thanks for this comment; we had erroneously labeled the figure description as describing “specific binding”; however, the curves represent *total* binding (i.e. specific as well as non-specific binding). In light of this correction, we hope the reviewer might appreciate that when total binding is considered, a 1-1 site binding model exhibits a sloped linear trend at saturation, as seen in the figure 2H.

- *Fig. 3B, C, F. These are LigandTracer binding plots. This technology is not widely used and the authors provide no interpretation of the plots, thus I would recommend deleting all of the LigandTracer plots in the paper. A saturation radioligand binding assay is the “gold standard” for evaluating K_D and B_{max} and is more readily appreciated by scientists in the field.*
- *Supplementary Fig. 3 should be deleted. There is no explanation of these LigandTracer binding curves to interpret the data, and as mentioned above, this is a non-standard method to evaluate binding affinity.*

Response: Upon considering the comments from reviewer #3, we have decided to omit the LigandTracer data in the revised manuscript.

- *Fig. 3D (Table). K_D values should be given to 1-decimal place. Also, there appears to be a major discrepancy between the B_{max} values which are “ $\times 10e4$ ” and the HER2s per cell which are “ $\times 10e5$ ”. Is there an error here? Same for Supplementary Table 1 – there may be an error in the B_{max} units.*

Response: As suggested by the reviewer, the significance values of the K_D have now been revised. The column with B_{max} values was removed to avoid confusion or redundancy; the values indicate the same thing. – i.e. B_{max} (which was indicated in CPM) is just a measure of the saturable number of HER2s per cell, albeit in an arbitrary unit. Please see Fig. 2H and Supplementary Fig. 4C.

- *Fig. 4A. The images of the cells need to be magnified to appreciate differences in internalized HER2 in control vs. lovastatin treated cells.*

Response: Fig. 4A is now Fig. 2J. We have magnified areas of those images per the reviewer’s suggestion. Two additional supplementary videos of live cell imaging with Trastuzumab-ICG have also been included in the manuscript for the reviewers and readers of this article to appreciate differences in membrane-associated HER2 statin-treated versus control cells. Please see Supplementary videos 1 and 2.

- *Fig. 4E, F. These figures should be in color. Also, what is the significance of the two separate sections? Are they two sections from the same tumor xenograft, or sections from two different xenografts? Please state in the legend.*

Response: Please see previous comment related to B&W images. Images are two representative sections of two different xenografts; we have now included this information in the respective legend and this is now Fig. 3D.

- *Fig. 5A. It seems that the background radioactivity in the liver, blood pool and other normal organs is lower for the lovastatin-treated mice, while the tumor intensity is similar for control and lovastatin-treated mice. Improved imaging with lovastatin appears to be due mainly to lower background.*

Response: Figure 5A is now Figure 4A. We thank the reviewer for noticing this discrepancy and pointing it out for us to make a correction. The images were originally presented using different scaling parameters. The control group was scaled for display between 0-10%, while the statin group was scaled for display between 0-20%. For easy comparison, we have synchronized the display of the PET images on a scale of 0-20 %ID/g. This should help overcome the previously existing discrepancy. Kindly see Fig. 4A. Background radioactivity is now in good agreement between the control versus statin-treated mice, as shown in both the images and BioD data.

- *Supplementary Fig. 4. Tumor intensity looks the same on the lovastatin-treated and control tumors in the mice on the coronal and transaxial sections. It is not clear that the lovastatin intratumoral injection increased the tumor uptake of ^{89}Zr -trastuzumab (at least on the images). Panels C and D are the tumor and normal tissue uptake, which do show a difference in tumor uptake with lovastatin treatment.*

Response: The images related to this preliminary approach of using an intratumoral administration of lovastatin were removed to avoid any misunderstandings. The BioD quantification shows that an intratumoral injection of lovastatin increases trastuzumab binding to tumors.

• *Supplementary Fig. 5. This shows the tumor and normal tissue biodistribution for control and lovastatin-treated mice. This is a key figure and should be in the main manuscript. There appears to be a modest increase in tumor uptake of ^{89}Zr -trastuzumab, but only slight differences in normal tissue uptake. However, this doesn't seem to agree with the images shown in Fig. 5A and Supplementary Fig. 5A, which show lower background in mice treated with lovastatin. The authors should explain this apparent discrepancy.*

Response: We agree with reviewer's comments that the biodistribution data constitutes a key figure to highlight our *in vivo* findings. These graphs have now been moved into the main manuscript. Kindly see Fig. 4B. Please see previous comment related with background radioactivity.

• *Fig. 5G. I am not enthusiastic about including the pre-targeting approach in this paper, as it is a separate study that requires controls to demonstrate specificity as well as optimization of the sequence of the trastuzumab and 18F-labeled small molecule administration etc.*

• *P. 11. Pre-targeting results. As mentioned above, I am not that enthusiastic to include the pre-targeting experiments in this paper.*

• *Supplementary Fig. 7. There is tumor imaging with the pre-targeting approach, but increased tumor uptake in the presence of lovastatin is still low (4% vs. 2% ID/g). Tumor:blood ratios are only slightly greater than 1 with lovastatin and much less than 1 without lovastatin. These are actually very low for a pre-targeting approach, which is normally intended to yield very high tumor:blood ratios. Also, at this early imaging time point (<4 h), it is not clear what proportion of tumor localization is due to specific binding to trastuzumab pre-bound to HER2 in the tumor, or due to tumor blood flow.*

Response: We agree with reviewer's comments that the pretargeting studies should belong to a different paper. We have removed those studies from this manuscript.

• *Fig. 5I. This study does not show an ADCC effect as the mice are immunocompromised. It likely shows an increased action of trastuzumab by other mechanisms such as inhibition of tumor growth signaling. There needs to be more details in the legend of this figure to indicate the type of mice, tumor xenograft, dose administration etc.*

• *P. 12. The authors need to point out that the *in vivo* effect of trastuzumab in mice with/without lovastatin is not due to ADCC since immunocompromised mice are used, and also the human Fc-domain of trastuzumab may not activate ADCC in mice.*

Response: The reviewer makes a very good point here. Although immunodeficient, nude mice have B-cells, functional NK-cells, functional macrophages and are capable of cytokine signaling. Therefore, while it may not be the predominant mechanism of action, the possibility of an ADCC effect may not completely be ignored in this mouse strain. That being said, we completely agree with the reviewer's point, and thus do not necessarily state ADCC as the sole mechanism of action responsible for the therapeutic effect observed in our experiments. Instead, in the revised manuscript we clearly discuss that results from the therapy studies using a combination of trastuzumab and lovastatin improved the therapeutic efficacy via temporal modulation of CAV1, which improves HER2 stability at the cell membrane and increases the avidity for trastuzumab binding to HER2-positive tumors *in vivo*. Ultimately, we believe that the remarkably enhanced uptake of trastuzumab in HER2-positive tumors treated with lovastatin potentiates the improvement in therapeutic responses through one of many plausible mechanisms of action of trastuzumab besides ADCC.

We agree with reviewer that other mechanisms might change after CAV1 depletion. A sentence has been added in the discussion section of the revised manuscript to clarify that ADCC is not a predominant mechanism of action contributing to the therapeutic efficacy of trastuzumab and/or combination of trastuzumab + lovastatin in our studies – mainly because these experiments were performed using immunodeficient mice. Please see page 14.

Page 14: "Admittedly, despite the successful *in vitro* demonstration of an effective trastuzumab-mediated ADCC potentiated by treatment with lovastatin, we realize that

ADCC may not be a predominant mechanism of action contributing to the therapeutic efficacy of trastuzumab and/or combination of trastuzumab with lovastatin in our preclinical *in vivo* studies, since these experiments were performed using immunodeficient mice.”

Additional information was added into the legend of Fig. 4 to include details about type of mice, tumor xenografts and dose of administration. Kindly, see page 34:

Page 32: “D, superior *in vivo* therapeutic efficacy of trastuzumab combined with lovastatin when compared with trastuzumab in nu/nu female mice bearing NCIN87 gastric and BT474 breast xenografts, and NSG mice bearing gastric PDXs. Intraperitoneal trastuzumab administration 5 mg/kg weekly (during 5 weeks) was started at day 0. Lovastatin (4.15 mg/kg of mice) was orally administrated 12 h prior and at the same time as the intraperitoneal injection of trastuzumab.”

- *P. 14. The authors should point out that interfering with HER2 internalization may decrease the effectiveness of trastuzumab-emtansine, a new drug used for HER2-positive breast cancer.*

Response: Excellent point.

This information is highlighted in discussion section. Kindly see page 14:

Page 14: “Receptor-mediated endocytosis of ADCs such as TDM1 is essential for the therapeutic efficacy of this class of drugs. Indeed, the internalization of ADCs precedes the release of the cytotoxic drug within the lysosomal compartment of the cell, leading to tumor regression.”

“In light of these reports, it would seem counter-productive to prevent HER2 internalization via pharmacological depletion of CAV1, as this might reduce the therapeutic benefit of HER2-directed ADCs.”

- *P. 14. The statement that HER2 internalization is not required for the therapeutic activity of trastuzumab is not correct, since one proposed mechanism of action of the drug is promotion of HER2 internalization, thereby decreasing HER2 on the cell surface, and thus decreasing HER2-mediated growth signaling.*

Response: We thank the reviewer for pointing this out. We have corrected this information in the discussion section of the revised manuscript. Please see page 14:

Page 14: “However, our data suggests that the temporal modulation of CAV1 sufficiently stabilizes HER2 and enriches target density on the tumor cell surface to enhance the tumor’s avidity for trastuzumab. The presence of such a target-rich sink could ultimately drive the *in vivo* pharmacokinetics of trastuzumab in favor of target-mediated drug disposition leading to a significantly enhanced accumulation of trastuzumab in the tumor at early time points. Being a transient and temporally controlled phenomenon that occurs under the influence of a pharmacological modulator of CAV1, normalization of cellular distribution of HER2 including receptor-mediated internalization of the antibody bound to it would occur once the transient *in vivo* effect of the CAV1 modulator has worn out.”

- *P. 14. The statement that ADCC is the major mechanism of action of trastuzumab is not correct. There are many different mechanisms of action proposed for trastuzumab, including inhibition of growth signalling, promotion of HER2 internalization, decreased angiogenesis, stabilization of the ECD of HER2 avoiding a truncated constitutively activated receptor etc.*

Response: We agree with the reviewer’s suggestion. Different sections of the manuscript have been changed to avoid any misunderstandings. Kindly see the following pages:

Page 4: “Clinically, the antitumor activity of trastuzumab is attributed to more than a single mechanism of action. Direct action of the antibody drug is premised on receptor downregulation and subsequent alterations to intracellular signaling including attenuation

of downstream pro-tumorigenic cell signaling, inhibition of HER2 shedding, and inhibition of tumor angiogenesis. On the other hand, indirect action due to activation of an immune response via antibody dependent cell-mediated cytotoxicity (ADCC) has also been proposed as a mechanism of action for this drug [15-17].”

• *P. 15. The statement that it takes a few days to weeks until images with good contrast are obtained with radiolabeled antibodies is absolutely incorrect. Good quality images can be obtained as early as 48 h after administration of radiolabeled intact IgG forms (sometimes as early as 24 h). Radiolabeled antibody fragments (Fab or F(ab')₂) clear from the blood and normal tissues quickly, and good quality images can be obtained as early as 6 h and certainly within 12-24 h.*

Response: We thank the reviewer for making this point. Although we agree that tumor delineation on clinical immunoPET is achievable by 24 h, high contrast images are only achieved at or after 72 h p.i. (please see reference 65). We have modified the writing and clarified this in discussion section. Please see page 16:

Page 16: “From the perspective of immunoPET, our study presents a major advance in the field of molecular imaging. Traditionally, the sluggish *in vivo* pharmacokinetics of full-length antibodies has necessitated their radiolabeling with long-lived positron-emitting radionuclides such as zirconium-89 ($t_{1/2} = 78.2$ h) to yield high contrast images between 72-120 h post-injection of the radioimmunoconjugate tracer. However, this manifests as a logistical inconvenience requiring patients to return to the clinic for a PET scan 4-5 days after they have been injected with the tracer [65]. However, we were able to obtain high contrast PET images as early as 4 h post-injection of ⁸⁹Zr-radiolabeled trastuzumab in HER2-positive tumors wherein CAV1 was pharmacologically modulated (Fig. 4). Such an accelerated accumulation of ⁸⁹Zr-labeled trastuzumab is of tremendous clinical relevance from the standpoint of being able to overcome the current pharmacokinetic limitation of full-length antibodies to achieve same- to next-day immunoPET imaging in patients.”

• *P. 16. As mentioned previously, I would not include the pre-targeting approach.*

Response: Pre-targeting experiments and results thereof have been removed from the manuscript.

• *P. 24. I think the dose of lovastatin is higher than the normal human dose. The human dose is 10-80 mg/day which corresponds to 0.14-1.4 mg/kg/day based on a 70 kg human, whereas the authors administered 4.15 mg/kg to mice.*

Response: Good question. The same question was raised by reviewer #1 and has been addressed.

Although the IC₅₀ of lovastatin for HMGCR is ~3-4nM, several *in vitro* studies aiming to deplete cholesterol have reported the use of lovastatin in the μM range (please see the following references: Svelto et al. *Am J Physiol Renal Physiol* 298: F266–F278, 2010; Huttner et al. *Traffic* 2000 1: 952–962; Chattopadhyay et al. *Scientific Reports* volume 7, Article number: 4484 2017).

The *in vitro* dose of lovastatin used in our experiments was based on previous reports that demonstrated 25 μM as the pharmacologic dose capable of depleting caveolin-1 protein. Kindly see page 9 and reference 40:

Page 9: “Previous studies have reported that 25 μM of lovastatin reduces the protein expression of CAV1 *in vitro* [40].”

To determine the *in vivo* dose of lovastatin, the following rationale was taken into account: Assuming that 5% lovastatin is bioavailable after oral administration, an intratumoral dose of 0.44 mg/kg of mice was deduced as 5% of an oral dose – i.e. 8.3 mg/kg of mice. Kindly see page 24:

Page 24: “The doses of lovastatin used in our study were lower when compared with the human maximum dose (80 mg/day): the 0.44 mg/kg and 8.3 mg/kg of mice doses

correspond to daily human doses of 2.15 and 40.14 mg assuming an average human body weight of 60 kg. It should be noted that the intratumoral dose 0.44 mg/kg of mice was defined as 5% of oral dose 8.3 mg/kg of mice, assuming 5% lovastatin bioavailability after oral administration.”

The maximum human dose of lovastatin is 80 mg/day, which corresponds to 1.33 mg/kg/day based on a 60 kg body weight. For conversion of the human dose to a mouse dose as *per FDA guidelines*, this dose is adjusted for differences in metabolic rates between man and mouse based on body surface area. Kindly see Nair and Jacob, Journal of Basic and Clinical Pharmacy, 2016, 7. The doses of lovastatin used in both imaging and therapeutic studies - 8.33 mg/kg and 4.15 mg/kg correspond to human equivalent doses of 0.68 and 0.34 mg/kg. Both of these are remarkably lower than the maximum human dose. Notably, patients use lovastatin on a daily basis, while in our study lovastatin was administrated twice a week.

Also, previous preclinical studies have demonstrated the therapeutic potential of lovastatin *per se* using small animal doses between 5 to 10 mg/kg. Changes have been made to the text to ensure clarity and avoid misunderstandings. Kindly see page 15:

Page 15: “The maximum human dose of lovastatin is 80 mg/day (1.33 mg/kg/day for a 60 kg adult). The doses of lovastatin used in our imaging and therapy studies – 8.33 and 4.15 mg/kg – correspond to human equivalent doses of 0.68 and 0.34 mg/kg, which are significantly lower than the maximum recommended daily dose in humans. Previous preclinical studies have demonstrated the therapeutic potential of lovastatin *per se* using doses between 5 to 10 mg/kg [56].”

Minor Points

- P. 4. References 7,14 pertain to PET imaging with ^{89}Zr -trastuzumab and do not include imaging with ^{89}Zr -trastuzumab emtansine.
- P. 4. Statement that “antibodies are cell membrane impermeable” is not quite accurate, since there is receptor-mediated internalization for some antibodies.
- P.4. Statement that current testing for HER2 evaluates overall HER2 expression (cell surface and intracellular) is not correct. The criteria for HER2 scoring relates to the intensity and proportion of cells with cell surface immunohistochemical staining or *in situ* hybridization techniques (e.g. FISH) to assess HER2 gene amplification.
- P. 5 line 135. “de” should be “the”.
- Fig. 1-I. The half-lives should be given in whole numbers or maximum one decimal place.
- Fig. 2C –The effect of CAV-1 siRNA knock-down only slightly increases HER2 on the cell membrane (i.e. in the presence of CHX to inhibit protein synthesis).
- Fig. 3D – This figure is not needed, and should just be explained in the text.
- Fig 4C is not needed. Please just explain in the methods section of the text.
- Fig. 5F is not needed. Please just explain in the methods section of the text.
- P. 14. Some references for trastuzumab-mediated radioimmunotherapy are needed. Also, there are examples of where interfering with HER2 internalization could decrease the effectiveness of radioimmunotherapy, for example some papers that have been published using low energy Auger electron emitters, such as ^{111}In .

Response: All the minor points were accepted and included in the revised version of the manuscript.

Reviewers' Comments:

Reviewer #1:

Remarks to the Author:

"We are currently investigating if lovastatin treatment alters HER2 interaction with other members of the HER family...."

The potential involvement of other receptor family members should be mentioned/discussed.

"Assuming that 5% lovastatin is bioavailable after oral administration"

This assumption must be justified with a relevant citation.

Inclusion of the new data indicating a 'rescue' by mevalonate supports the author's conclusions.

Other concerns from this reviewer have been satisfactorily addressed. However, there may still be concerns from reviewers 2 and 3.

Reviewer #2:

Remarks to the Author:

Review of Pereira et al.,
NCOMMS-18-11045A

The revised manuscript is much improved. Well done! Congratulations to the authors for their effort and responses. The additional xenograft data (Fig 4D) are impressive.

One significant issue remains concerning the author's conclusions. The take home message of the manuscript is that CAV1 directs the trafficking of HER2 and this trafficking affects the potency of trastuzumab. When CAV1 expression is silenced there is very little change in HER2 expression (Fig 2A, 2H). Conversely, when CAV1 is overexpressed the localization of HER2 at the cell surface is profoundly altered (Fig 2B).

I do not buy the authors conclusions that the modest, if any, increase HER2 accumulation explains their data. As stated above, I believe that their data indicates that CAV1 depletion alters HER2 and HER2-trastuzumab trafficking. Most importantly, the authors do not need to make this claim because their data are compelling and the alteration in trafficking explains it.

Thus, the following lines in their manuscript should be changed to reflect that this is the case or at least tone down (as they did in lines 203-205) their assertion that HER2 levels are increased:

Lines: 73; 225 (note this line describes data in 2G where radioactivity not number of binding sites is being measured); 230; 234 (do the authors really think that an increase from 1,570,000 HER2s to 2,200,000 is really going to make a biological difference in the growth and survival signals emanating from constitutively activated HER2?); 258-259; 286 if the authors want to make the 'avidity' claim they should perform binding studies with a Fab version of trastuzumab and compare the results to the intact antibody. 324; 340; 382-383 'substantial'? Data do not support this claim.; 428-429 same old same old.

Specific Comments:

1. Fig 1A- Western blots of HER2 and CAV1 should be shown in Supplementary Figures.

2. Fig 1E- CAV1 western blot is very poor quality
3. Fig 1G- correct significant figures for error
4. Fig. 2A- Not clear that HER2 expression has increased with CAV1 KD. Have the authors performed densitometry on these blots? Do they have replicates?
5. Fig 2D- significant figures in error
6. Fig 2G- do not agree that the avidity of trastuzumab has increased given the magnitude of the difference.
7. Fig 2H- How many times was this experiment repeated? N87 cells have 1-2,000,000 HER2 per cell (not 200,000 as reported in 2H). Did the authors drop an order of magnitude in their calculations?
8. Fig 3D-3F- where are the statistics and number of replicates that back up the claim that the difference is significant?

Reviewer #3:

Remarks to the Author:

This is a revised version of the paper entitled "Caveolin-1 Mediates Cellular Distribution of HER2 and Affects Trastuzumab Binding and Therapeutic Efficacy". The authors have made significant revisions in the text and figures of the manuscript to address the reviewers comments and have provided a detailed point-by-point response to these comments. I am very satisfied with the revisions to the manuscript and responses to my comments in the review of the paper. I believe that the imaging sections of the paper are now greatly improved and more clear. I especially appreciate the adjustment of the images to make the background and tumour uptake comparable for Fig. 4A and the inclusion of the biodistribution data (Fig. 4B) in the main manuscript. The authors have removed the pre-targeting study, which I agree makes the paper more focused on the main hypothesis, that caveolin-1 controls the expression of HER2 and interventions to modulate caveolin-1 effect imaging with 89Zr-labeled trastuzumab and treatment with trastuzumab in HER2-positive tumours. The paper is a very through study which makes a very interesting and novel contribution to the literature on targeting HER2 for imaging and therapy of cancer.

I only have two remaining minor points that I would like the authors should consider:

1. Fig. 2H - This shows a binding curve for the binding of 89Zr-trastuzumab to NC1NH87 cells. As mentioned in the previous review, this curve does not show saturation, which may make the Bmax values inaccurate. In the response to this point, the authors state that the curve only shows the total binding and not the specific binding. Total binding would not be expected to saturate while specific binding would be expected to show saturation. It seems that the authors did not do a binding curve in the presence of an excess of unlabeled trastuzumab to generate the non-specific binding curve, which could then be subtracted from the total binding curve to generate the specific binding curve. This is not ideal. However, if this is the case, the authors should be able to generate the specific binding curve from the total binding curve based on the shape of the total binding curve. There is an algorithm to do this in GraphPad Prism software. I would recommend that they employ this algorithm to generate and plot the specific binding curves for Fig. 2H and also Suppl. Fig. 4C.

2. Response to Comment on p. 15 (page 17/18 in response document): "please see reference 65". There are only 62 references in the revised paper - are there some missing references?

Reviewers' comments:

Reviewer #1 (Remarks to the Author):

“We are currently investigating if lovastatin treatment alters HER2 interaction with other members of the HER family....”

The potential involvement of other receptor family members should be mentioned/discussed.

Response: Taking this comment into consideration, we have included the following sentence in the discussion section:

Page 12, “Further studies are necessary to evaluate the consequences of lovastatin treatment on the dimerization status of HER2.”

“Assuming that 5% lovastatin is bioavailable after oral administration”

This assumption must be justified with a relevant citation.

Response: We have added a new reference to support this assumption. Please see reference 57 in the new version of our manuscript.

Inclusion of the new data indicating a ‘rescue’ by mevalonate supports the author’s conclusions.

Other concerns from this reviewer have been satisfactorily addressed. However, there may still be concerns from reviewers 2 and 3.

Response: We thank the reviewer for suggesting the mevalonate experiments. We believe that the suggested experiments have highly increased the quality of our manuscript and strengthened the conclusions of our study.

Reviewer #2 (Remarks to the Author):

The revised manuscript is much improved. Well done! Congratulations to the authors for their effort and responses. The additional xenograft data (Fig 4D) are impressive.

Response: We appreciate the reviewer’s positive response about the revised manuscript and for providing additional feedback and suggestions.

One significant issue remains concerning the author’s conclusions. The take home message of the manuscript is that CAV1 directs the trafficking of HER2 and this trafficking affects the potency of trastuzumab. When CAV1 expression is silenced there is very little change in HER2 expression (Fig 2A, 2H). Conversely, when CAV1

is overexpressed the localization of HER2 at the cell surface is profoundly altered (Fig 2B).

I do not buy the authors conclusions that the modest, if any, increase HER2 accumulation explains their data. As stated above, I believe that their data indicates that CAV1 depletion alters HER2 and HER2-trastuzumab trafficking. Most importantly, the authors do not need to make this claim because their data are compelling and the alteration in trafficking explains it.

Response: Thank you for the feedback and comments. We agree that changes in HER2 protein levels at the cell membrane (as determined by cell surface biotinylation assays) after the knockdown of CAV1 via siRNA appear subtle (Fig 2A) compared to the dramatic alteration in cell surface HER2 protein levels when CAV1 is overexpressed using CRISPR activation plasmid (Fig 2B). Nevertheless, an ImageJ-based densitometric evaluation of the western blot shown in Fig 2A yielded a 1.4 ± 0.2 -fold increase in HER2 protein at the membrane of cells treated with CAV1-siRNA compared to those treated with a scrambled siRNA. This quantitative information is now included in the revised version of Legend of Figure 2 and described in the main manuscript. Kindly see page 6:

Page 6: “siRNA-mediated knockdown of CAV1 slightly increased (1.4 ± 0.2 , mean \pm S.E.M, $n = 3$) HER2 at the cell membrane for NCIN87 cells (Fig. 2A)...”

Notably, our attempts to knockout CAV1 using the CRISPR/Cas9 KO plasmid system resulted in cell death. So, we resorted to the use of the siRNA platform to knockdown CAV1. Unlike complete silencing of genes that can be achieved in a CRISPR/Cas9-generated knockout, the siRNA platform used in our study is able to knockdown protein expression (Fig. SI3A) in a transient manner.

Nonetheless, CAV1 depletion via siRNA was found to increase the half-life of HER2 at the cell surface in different cancer cell lines (Fig. 2D, Fig. SI3D-F), resulting in an increase in the number of HER2 receptors (B_{\max}) present at a given time on the surface of tumor cells.

Admittedly, the incomplete and transient knockdown of CAV1 expression via siRNA (Fig. SI3A, Fig. 2A) might be a strong reason that precludes the visualization of a high increase in the half-life and localization of HER2 at the cell membrane. Instead, a better appreciation of the increase in membrane-localized HER2 upon CAV1 depletion can be made from *in vivo* experiments wherein lovastatin was used as the pharmacological modulator of membrane cholesterol and CAV1. All the results shown in figure 3 strongly support this conclusion.

Furthermore, appended below is a western blot analysis of biotinylated cell surface-associated HER2 in lysates of NCIN87 s.c. tumors from athymic nude mice (n=2 per group) that were euthanized at different time points t = 0, 12, 18, 24 h after treatment with lovastatin.

This image has now been included in the supplementary information as Fig. S5. We contend that a better visualization of the increase in protein levels of HER2 at the tumor cell membrane in relation to the depletion of CAV1 is better realized *in vivo* using a pharmacological modulator of CAV1 instead of siRNA-based knockdown.

In sum, what appears to be a modest increase in the localization of HER2 at the membrane – owing to the limited *in vitro* depletion of CAV1 using the siRNA technology – is actually a highly pronounced phenomenon when analyzed using lovastatin as an *in vivo* pharmacological modulator of CAV1.

Finally, we find that the reviewer’s suggested take-home message “*CAV1 depletion alters HER2 and HER2-trastuzumab trafficking*” resonates extremely well with the title of our manuscript “*Caveolin-1 Mediates Cellular Distribution of HER2 and Affects Trastuzumab Binding and Therapeutic efficacy*”.

Admittedly, we did not intend to specifically determine how CAV1 depletion alters the trafficking or internalization of HER2 and/or HER2-trastuzumab immunocomplexes. We believe that making such a conclusion will require the use of more comprehensive methodology to investigate the dynamics of the endocytic pathway in the context of HER2 and/or HER2-trastuzumab (e.g. determination of HER2 or HER2-trastuzumab in specific trafficking vesicles and so on). Such experiments that may be necessary to make a conclusion in the context of intracellular trafficking were not the scope of the present manuscript.

Thus, the following lines in their manuscript should be changed to reflect that this is the case or at least tone down (as they did in lines 203-205) their assertion that HER2 levels are increased:

Response: The manuscript has been carefully reviewed to ensure clarity and avoid any misunderstanding. We have replaced “increase in membrane HER2” by “increase in HER2 half-life and availability at the cell membrane”. Changes have been made in all the lines as suggested by the reviewer.

Lines: 73;

Response: Our paragraph was modified to: “...depletion with lovastatin increases HER2 half-life and availability at the cell membrane...”

Line 225 (note this line describes data in 2G where radioactivity not number of binding sites is being measured);

Response: Our paragraph was modified to: “...treated with ⁸⁹Zr-labeled trastuzumab revealed a significant increase (P < 0.001) in membrane-associated radioactivity...”

230; 234 (do the authors really think that an increase from 1,570,000 HER2s to 2,200,000 is really going to make a biological difference in the growth and survival signals emanating from constitutively activated HER2?);

Response: The number of receptors calculated in our experiment (B_{max}) is a measure of membrane receptors and not total receptors. We do believe that this increase in the number of receptors combined with an increment in HER2 membrane half-life justifies the increase of trastuzumab binding to the cell surface of tumor cells.

In our experiments, we did not detect alterations in pHER2 (Fig. 2I) upon CAV1 depletion with lovastatin.

Further experiments are necessary to understand whether or not a lovastatin treatment “*makes a biological difference in the growth and survival signals emanating from constitutively activated HER2*”. Such experiments were not the scope of the present manuscript. Further studies are necessary to determine the mechanisms by which combination of lovastatin with trastuzumab improves therapy in HER2⁺/CAV1⁺ tumors.

258-259; 286 if the authors want to make the ‘avidity’ claim they should perform binding studies with a Fab version of trastuzumab and compare the results to the intact antibody. 324;

Response: We thank the reviewer for a careful reading of the revised manuscript and for pointing out our mistake with the language here. We would

like to make a clarification. We meant to suggest that *the increase in half-life of HER2 at the tumor cell surface upon lovastatin treatment enhances the target density/bioavailability in the tumors, causing them to become highly avid for anti-HER2 antibodies – including ⁸⁹Zr-labeled Trastuzumab.*

The word avidity was used in the context that variations in the levels of receptor (in our case, HER2) at the cell membrane affect *the avidity of the tumor* for the tracer in the study. Thus, while there is practically no alteration in the binding affinity of Trastuzumab for HER2, we believe that an increase in tumor avidity in mice treated with lovastatin arises as a consequence of higher/enhanced membrane density/bioavailability of HER2. We intended to use the term “avidity” in the context of PET tracers. For example, some tumors are more “FDG-avid” than others – meaning to suggest that some tumors have higher glycolytic activity than others.

We have addressed this mistake by making appropriate changes in the revised manuscript to clarify our stance and would like to avoid any further misunderstanding in this regard.

340;

Response: The claim that “...increases trastuzumab binding to HER2-positive tumors.” is well supported in our *in vitro* (Bmax, cell fractionation) and *in vivo* biodistribution studies with radiolabeled trastuzumab.

382-383 ‘substantial’? Data do not support this claim.; 428-429 same old same old.

Response: The paragraph 382-383 was modified to: “that CAV1 knockdown reduces HER2 internalization, resulting in an increase of HER2 half-life at the cell membrane...”

The paragraph 428-429 was modified to: “However, our data suggests that the temporal modulation of CAV1 increases HER2 half-life at the tumor cell surface to enhance the tumor’s avidity for trastuzumab.”

Specific Comments:

1. *Fig 1A- Western blots of HER2 and CAV1 should be shown in Supplementary Figures.*

Response: We have now included Western blots of HER2 and CAV1 on the cell lines analyzed in our study. Please see Supplementary Fig. 1.

2. *Fig 1E- CAV1 western blot is very poor quality*

Response: Isolation and further fractionation (in our case, a total of 10 membrane fractions) of cell membrane proteins is a protocol with inherent difficulties that explain the quality of the image shown in Fig. 1E. We are hoping that the printed journal version of this figure will be able to better reproduce the visual gains.

3. *Fig 1G- correct significant figures for error*

Response: Thanks for this comment; we have corrected the image for error.

4. *Fig. 2A- Not clear that HER2 expression has increased with CAV1 KD. Have the authors performed densitometry on these blots? Do they have replicates?*

Response: We have now included densitometric quantification of HER2 expression in the membrane fraction of the WB shown in Fig. 2A. CAV1 depletion increases 1.4 HER2 protein levels at the cell membrane of tumors cells. Please see page 6:

“siRNA-mediated knockdown of CAV1 slightly increased (1.4 ± 0.2 , mean \pm S.E.M, $n = 3$) HER2 at the cell membrane for NCIN87 cells (Fig. 2A).”

5. *Fig 2D- significant figures in error*

Response: Thanks for this comment; we have corrected the image for error.

6. *Fig 2G- do not agree that the avidity of trastuzumab has increased given the magnitude of the difference.*

Response: Please see our responses to the previous comments related to the increase in HER2 half-life and availability at the cell membrane and the aspect of avidity. We contend that the increase in HER2 half-life and its bioavailability at the membrane are better appreciated in experiments where lovastatin was used as a pharmacological modulator of CAV1 unlike the *in vitro* CAV1 knockdown experiments. In light of the fact that the magnitude of difference in the membrane-localized HER2 is significantly high upon CAV1 modulation by the statin, this makes the tumors highly avid for trastuzumab. This is corroborated by all the data shown figure 3 – specifically, by the dramatic increase in the immunofluorescence staining of HER2 at the cell membrane of tumors that were pretreated with lovastatin and the increase in membrane-bound ^{89}Zr -trastuzumab of cells pre-treated with lovastatin. Kindly note that the term “avidity” was used to indicate the *increase in the tumor’s avidity for trastuzumab*.

7. *Fig 2H- How many times was this experiment repeated? N87 cells have 1-2,000,000 HER2 per cell (not 200,000 as reported in 2H). Did the authors drop an order of magnitude in their calculations?*

Response: We have performed 3 independent experiments. The previous references that we found reporting the number of HER2 receptors in NCIN87 gastric cancer cells describe a value of 800,000 receptors per cell.

Bergstrom DA, et al. (2015) A novel, highly potent HER2-targeted antibody-drug conjugate (ADC) for the treatment of low HER2-expressing tumors and combination with trastuzumab-based regimens in HER2-driven tumors. [abstract]. In: Proceedings of the 106th Annual Meeting of the American Association for Cancer Research; 2015; Philadelphia, PA. Philadelphia (PA): AACR; Cancer Res 75(15 Suppl):Abstract nr LB-231

Innovations for next generation antibody-drug conjugates. 2018. Page 229.

We recall that the number of HER2 receptors can be different using different methodologies. In our approach, we have tried different previously reported methodologies to determine B_{max} values. Unfortunately, we were not able to reproduce some of those previous reported results because of lack of information in the methods section related to for example time of incubation, temperature and concentrations. In our experiments, the number of HER2 receptors was obtained with a saturation-binding assay at 4C using ^{89}Zr -labeled trastuzumab.

8. *Fig 3D-3F- where are the statistics and number of replicates that back up the claim that the difference is significant?*

Response: Excellent point. Immunofluorescence assays shown in Fig 3D,F were used as qualitative readouts in our study. The word “significant” was removed to avoid any misunderstandings.

Reviewer #3 (Remarks to the Author):

This is a revised version of the paper entitled "Caveolin-1 Mediates Cellular Distribution of HER2 and Affects Trastuzumab Binding and Therapeutic Efficacy". The authors have made significant revisions in the text and figures of the manuscript to address the reviewers' comments and have provided a detailed point-by-point response to these comments. I am very satisfied with the revisions to the manuscript and responses to my comments in the review of the paper. I believe that the imaging sections of the paper are now greatly improved and more clear. I especially appreciate the adjustment of the images to make the background and tumour uptake comparable for Fig. 4A and the inclusion of the biodistribution data (Fig. 4B) in the

main manuscript. The authors have removed the pre-targeting study, which I agree makes the paper more focused on the main hypothesis, that caveolin-1 controls the expression of HER2 and interventions to modulate caveolin-1 effect imaging with ⁸⁹Zr-labeled trastuzumab and treatment with trastuzumab in HER2-positive tumours. The paper is a very thorough study, which makes a very interesting and novel contribution to the literature on targeting HER2 for imaging and therapy of cancer.

Response: Thank you for taking the time to review the revised version of our manuscript. We have addressed the remaining comments, as described below.

I only have two remaining minor points that I would like the authors should consider:

1. Fig. 2H - This shows a binding curve for the binding of ⁸⁹Zr-trastuzumab to NCINH87 cells. As mentioned in the previous review, this curve does not show saturation, which may make the Bmax values inaccurate. In the response to this point, the authors state that the curve only shows the total binding and not the specific binding. Total binding would not be expected to saturate while specific binding would be expected to show saturation. It seems that the authors did not do a binding curve in the presence of an excess of unlabeled trastuzumab to generate the non-specific binding curve, which could then be subtracted from the total binding curve to generate the specific binding curve. This is not ideal. However, if this is the case, the authors should be able to generate the specific binding curve from the total binding curve based on the shape of the total binding curve. There is an algorithm to do this in GraphPad Prism software. I would recommend that they employ this algorithm to generate and plot the specific binding curves for Fig. 2H and also Suppl. Fig. 4C.

Response: We agree with the reviewer's suggestion. We have corrected Fig. 2H and Supplementary Fig. 4C to include the graph with the specific binding curve.

2. Response to Comment on p. 15 (page 17/18 in response document): "please see reference 65". There are only 62 references in the revised paper - are there some missing references?

Response: We thank the reviewer for pointing this out. Reference 65 corresponds to reference 6 in our manuscript. We had erroneously labeled

Reviewers' Comments:

Reviewer #2:

Remarks to the Author:

The revised manuscript addresses my issues and is now acceptable. CONGRATULATIONS to the authors on a well-executed and well-written study.

Reviewer #3:

Remarks to the Author:

The authors have very well addressed the two final comments that I provided on the last review. I have no further comments. The authors are to be congratulated on a very interesting, thorough and novel study, that I believe will be of great interest to the molecular imaging community as well as the experimental cancer therapeutics community.

Reviewers' comments:

Reviewer #2 (Remarks to the Author):

The revised manuscript addresses my issues and is now acceptable. CONGRATULATIONS to the authors on a well-executed and well-written study.

Response: We are very grateful to the reviewer for the positive overview of the manuscript. We thank again the reviewer for the suggestions to improve the quality of the manuscript. We believe that, thanks to reviewer's precious inputs, the manuscript was highly improved.

Reviewer #3 (Remarks to the Author):

The authors have very well addressed the two final comments that I provided on the last review. I have no further comments. The authors are to be congratulated on a very interesting, thorough and novel study, that I believe will be of great interest to the molecular imaging community as well as the experimental cancer therapeutics community.

Response: We are very grateful to the reviewer for the positive overview of the manuscript. We thank reviewer's suggestions, which have highly increased the quality of our work.